# Algorithmic Capabilities of Random Transformers

**Ziqian Zhong, Jacob Andreas**
Massachusetts Institute of Technology
{ziqianz, jda}@mit.edu

## Abstract

Trained transformer models have been found to implement interpretable procedures for tasks like arithmetic and associative recall, but little is understood about how the circuits that implement these procedures originate during training. To what extent do they depend on the supervisory signal provided to models, and to what extent are they attributable to behavior already present in models at the beginning of training? To investigate these questions, we investigate what functions can be learned by randomly initialized transformers in which only the embedding layers are optimized, so that the only input–output mappings learnable from data are those already implemented (up to a choice of encoding scheme) by the randomly initialized model. We find that these random transformers can perform a wide range of meaningful algorithmic tasks, including modular arithmetic, in-weights and in-context associative recall, decimal addition, parenthesis balancing, and even some aspects of natural language text generation. Our results indicate that some algorithmic capabilities are present in transformers (and accessible via appropriately structured inputs) even before these models are trained.[1]

## 1 Introduction

A large body of recent work has demonstrated the effectiveness of transformer language models (LMs) [46] on general sequence-modeling tasks. Transformers seem to be especially well-suited (relative to other flexible neural models) at problems involving numerical reasoning [41, 24, 30], string manipulation [28], and various forms of in-context learning [7, 17, 1, 25]. Why is this the case?

One possibility is that some aspect of the transformer architecture makes these behaviors easy to learn. Under this hypothesis, transformer models do not implement any useful functionality when initialized; however, their loss landscape is structured such that they can be (computation- and sample-) efficiently optimized for behaviors of interest. But another possibility is that—because of intrinsic properties of the transformer architecture and parameter initialization schemes—these capabilities are *already implemented* in some fashion even in randomly initialized models.

To disentangle these possibilities, we investigate the behavior of randomly initialized transformer models in which *only the embedding layers are optimized*, leaving all other model-internal parameters fixed. If such embedding-only training is successful, it implies that the randomly initialized model's behavior on some subspace already corresponds to the input–output mapping of interest, up to a choice of encoding scheme—in other words, that the randomly initialized model can already perform the target task, and it suffices to find an encoding of inputs and outputs that induces the target behavior.

In experiments on seven tasks, we find that embedding-only training yields accurate models for a diverse set of problems spanning arithmetic, associative recall, and sequence generation—in some cases substantially outperforming similarly trained recurrent models. Remarkably, transformer *language models* trained in this fashion can even produce grammatical (though largely nonsensical)

---

[1]Code is available at https://github.com/fjzzq2002/random_transformers.

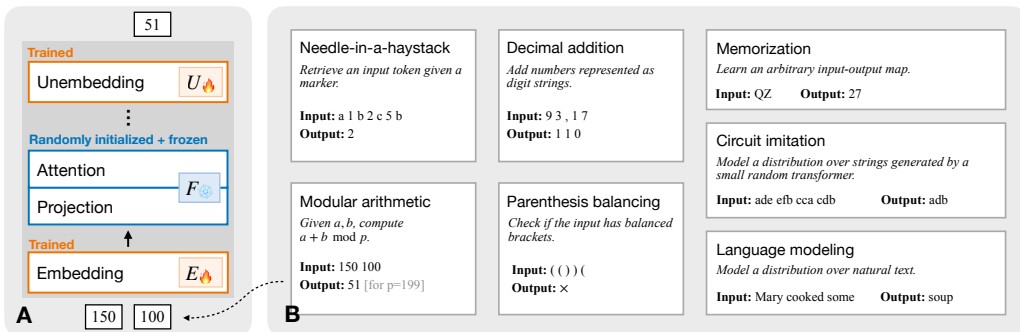

Figure 1: Overview of problem setup. **A**: Modeling approach. We initialize transformers randomly, then optimize *only* their input and output embedding layers on a dataset of interest. We find that these random transformers can be successfully trained to perform a diverse set of human-meaningful tasks. **B**: Task set. We evaluate the effectiveness of random transformers on a set of model problems involving arithmetic and memorization, as well as modeling of natural language text.

natural language text. We explain these results by showing that embedding-only training steers both inputs and model-internal representations into low-dimensional subspaces on which the model implements the target computation, a phenomenon we call "subspace selection". Embedding-only training is most successful when the target computation can be performed in a subspace that is low-dimensional relative to the ambient dimension of the model's hidden representations.

These findings build on a long line of research aimed at understanding the effectiveness of deep networks in terms of their behavior at initialization—e.g. showing that random convolutional networks are high-quality feature extractors [3, 8], or that overparameterized networks can be pruned down to sparse "lottery ticket" subnetworks that implement the correct behavior [16, 53, 39, 11]. But in contrast to past work, the solutions found by embedding-only training involve algorithmic computation rather than feature extraction, performed in low-dimensional subspaces but not by sparse sub-networks. Even more generally, our results show that pruning and optimization are not always necessary to surface useful capabilities—at least in transformers, some capabilities are available as soon as models are initialized, requiring only a learned encoding of inputs. This in turn suggests that it may be possible to partially understand the effectiveness of transformers simply by understanding their behavior at initialization.

Our work also has implications for research on circuit-level interpretability of transformers and other neural models: if even random models can perform structured, algorithmic tasks, then attempts to understand models by directly inspecting parameter matrices—and not their behavior on natural data distributions—may be fundamentally limited in their ability to characterize learned behaviors.

## 2 Background and Related Work

**Random feature extractors** Random deep convolutional networks are highly effective visual feature extractors even without training. Jarrett et al. [23] first discovered that linearly combining features from a randomly initialized one-layer convolutional network achieved comparable performance to fully trained networks for downstream vision tasks. Saxe et al. [40] showed that performance improvements from training is relatively minor comparing to architectural changes. In this work, we expanded the discussion to language models and demonstrated that training embeddings alone is sufficient to succeed in many tasks, highlighting the strong inductive bias of transformer architectures.

**Neural reprogramming** Neural reprogramming aims to repurpose existing neural networks for novel tasks via simple transformation layers. This technique was first purposed by Elsayed et al. [15] as a way to exploit trained neural network served by existing providers, and it was later used as a resource-efficient domain-transfer technique [45, 49]. In our work, we showed that in addition to neural networks trained for other tasks, even randomly initialized neural networks can be reprogrammed to achieve non-trivial performance.

**Sparse sub-networks and lottery tickets**  The lottery ticket hypothesis was first proposed by Frankle and Carbin [16]: a randomly-initialized dense neural network contains subnetworks, the *winning ticket*s, that when trained in isolation can match the performance of the original network. Zhou et al. [53] and Ramanujan et al. [39] strengthened the hypothesis by discovering pruned subnetworks that achieve comparable accuracy of the trained full network within untrained, randomly initialized networks - winning tickets do not even need to be trained. The hypothesis also holds in transformers [11, 44]. Similar to the hypothesis, our work also focuses on the models' capabilities at initialization, but we showed that these capabilities can be surfaced without any pruning.

**Interpreting neural circuits**  Many efforts have been dedicated to the interpretation of neural networks. On the arithmetic task we study in this paper, Nanda et al. [31] and Zhong et al. [52] described two different mechanisms in transformers on modular addition; Quirke and Barez [37] performed detailed mechanistic analysis for one-layer transformers trained on decimal addition. These studies provide insights into possible mechanisms transformers might employ to solve these tasks. With theoretical analysis, Wen et al. [47] proved that on the bounded Dyck task, the attention patterns and weights of neural circuits could be quite arbitrary, which we confirm in this work.

**Reservoir computing**  Reservoir computing is a framework for computation. In the diagram, a blackbox reservoir receives data and updates its inner states there upon. A simple readout mechanism is then trained to map its inner states to the desired output. The reservior is generally kept untrained and only the readout part is trained. Under our notations, we can interpret the paradigm as training only the unembedding part of random neural networks. See Benjamin et al. [42] for a general overview for reservoir computing and Mantas et al. [29] for a survey on its applications on recurrent neural networks.

## 3   Setup

**Models**  We study the behavior of decoder-only transformer language models. In these models, **inputs** $x$ (represented as sequence of token IDs) are first assigned vector **embeddings** $h^{(0)} = E(x)$ via an **embedding layer** $E$. These embeddings are then passed through a series of $m$ **intermediate layers** $F^{(1)}, F^{(2)}, \cdots, F^{(m)}$ so that $h^{(i)} = F^{(i)}(h^{(i-1)})$, with each $F^{(i)}(x)$ computing a **hidden representation** $h^{(i)}$ via a transformation:

$$h^{(i)} = F^{(i)}(h^{(i-1)}) = \text{FFN}^{(i)}(\text{SelfAtt}^{(i)}(h^{(i-1)})) \,,$$

where FFN, SelfAtt are feed-forward and causal self-attention modules as in Radford et al. [38] (layer norms are omitted for simplicity). The final activation $h^{(m)}$ is mapped by a **unembedding layer** $U$ to a distribution over next tokens.

To encode information about the ordering of input tokens, we implement the embedding layer $E$ using two matrices: a token embedding matrix $E_{\text{token}}$ and a positional embedding matrix $E_{\text{pos}}$. For an input $x = [x_1, x_2, \cdots, x_n]$, we first calculate the initial activations

$$h^{(0)} = E([x_1, x_2, \cdots, x_n]) = [E_{\text{token}}[x_1] + E_{\text{pos}}[1], E_{\text{token}}[x_2] + E_{\text{pos}}[2], \cdots, E_{\text{token}}[x_n] + E_{\text{pos}}[n]].$$

Similarly, the unembedding layer is parameterized by a single matrix, and model predictions have the form:

$$p(x_{n+1} \mid x_{1 \cdots n}; E, F, U) \triangleq \text{softmax}\left(U h_n^{(m)}\right)[x_{n+1}],$$

where (in a slight abuse of notation) $E$, $F$ and $U$ denote embedding, intermediate, and unembedding parameters respectively.

In terms of parameters, let the hidden dimension be $d$, the number of layers be $m$, the vocabulary size be $v$ and the maximum context length be $n$, the embedding layers have $\Omega((n+v)d)$ parameters as matrix $E_{\text{token}}$ and $U$ have shape $v \times d$ and matrix $E_{\text{pos}}$ has shape $n \times d$, while the full network has an extra $\Omega(md^2)$ parameters.

**Initialization**  Models are trained via gradient descent from some random initial parameterization. Following Radford et al. [38], parameters of feed-forward layers are initialized by sampling from isotropic Gaussians with mean 0 and standard deviation $0.02/\sqrt{2n}$. All the other weight matrices are initialized with 0-mean Gaussians with standard deviation 0.02. The affine transformations in layer normalizations are initialized as identity.

**Training**    In this work, we examine language models with frozen intermediate layers, which we call **random transformers**. In these models, we fix the randomly chosen parameters intermediate layers, and train only the embedding layer $E$ and unembedding layer $U$. Our experiments thus compare:

$$\textbf{Full Training:} \quad \underset{E,F,U}{\arg\min} \sum_{x,n\geq 0} -\log p(x_{n+1} \mid x_{1\cdots n}; E, F, U)$$

$$\textbf{Embedding-Only Training:} \quad \underset{E,U}{\arg\min} \sum_{x,n\geq 0} -\log p(x_{n+1} \mid x_{1\cdots n}; E, F, U)$$

where the $\arg\min$ is computed approximately via mini-batch stochastic gradient descent.

## 4    Random Transformers Can Perform Simple Algorithmic Tasks

Can random transformers be steered to perform meaningful tasks by optimizing only input and output tokens' embeddings? We begin by evaluating four widely-studied tasks that serve as toy models of important behaviors in large-scale LMs.

### 4.1    Tasks

**Modular Addition**    This task evaluates models' ability to perform integer addition under a fixed prime modulus $p = 199$. Models receive a sequence of input tokens $[a, b]$ for $a, b \in [0, p - 1]$ and must compute $(a + b) \bmod p$. When over-parameterized models are trained to perform this task, grokking (a long period of memorization followed by an abrupt transition to generalization [36]) is typically observed [26, 18]. Neural sequence models of different kinds have been found to implement two interpretable algorithms, sometimes referred to as the "Clock" [31] and "Pizza" [52], when trained to perform this task.

**Needle-in-a-Haystack**    This task evaluates models' abilities to process long input sequences [4]. In the variant we study, models receive as input a sequence of form $[m_1, c_1, m_2, c_2, \cdots, m_k, c_k, \underline{m_u}]$. Here, $m_1, m_2, \cdots, m_k$ are distinct *markers* ($k \leq 30$) and $c_i$'s are corresponding *values*. The input ends with a marker $\underline{m_u}$ ($u \in [1, k]$), and models must search for the previous occurrence of that marker in the input sequence and output the corresponding $c_u$. Specific circuits like *induction heads* are often observed in models that perform this task [34].

**Decimal Addition**    This task evaluates models' ability to perform arithmetic operations distributed over sequences of multiple input tokens—in this case, addition of two equal-length numbers represented as digit sequences in base 10. The order of digits of both numbers and the results are reversed to simplify the task. For example, the task 39+71=110 is encoded as a pair with input 9 3 1 7 and output 0 1 1. We use 10-digit numbers in our setup. Past work has found that fully trained models can reliably learn some versions of this task [32, 51, 43].

**Parenthesis Balancing (Dyck Recognition)**    In this task, models are presented with a sequence of parentheses, and must predict whether they are balanced—i.e., whether the sequence contain an equal number of opening and closing parentheses, and within every prefix, the number of closing parentheses is no greater than the opening parentheses. Such sequences are also called Dyck sequences, and have been widely studied in language models because of their connection to context-free models of natural language syntax [50, 47]. Note that this task has a vocabulary of size $4$ (two parentheses and two labels), so only a very small number of parameters are optimized by embedding-only training. In our setup, the input parenthesis sequences have lengths at most 60.

For the modular addition task, we partition the full set of well-formed input–output pairs into a fixed train/test split; for the other problems, we pre-generate a fixed test set but randomly generate new pairs for each training batch. Additional details may be found in Appendix D.1.

### 4.2    Results

Results are shown in Table 1. Here we compare random transformers with a hidden size of 1024 to fully trained models with hidden sizes of 16 and 1024. For reference, we also compare to a (fully

| Task | Random 1024 | Random 16 | Normal 1024 | Normal 16 | LSTM 1024 |
|------|-------------|-----------|-------------|-----------|-----------|
| Modular Addition | 100.0% | 1.3% | 100.0% | 97.2% | 100.0% |
| Needle-in-a-Haystack | 100.0% | 7.5% | 100.0% | 12.0% | 99.5% |
| Decimal Addition | 100.0% | 26.6% | 100.0% | 67.5% | 53.0% |
| Parenthesis Balancing | 100.0% | 87.3% | 92.3% | 100.0% | 100.0% |

Table 1: Test accuracy of fully trained and random transformers, as well as fully trained LSTMs, on algorithmic tasks. Denoted numbers (1024 and 16) are hidden sizes; results are median over 10 random restarts. Random models with only trained embeddings reliably perform all four tasks, and even outperform fully trained LSTMs. See Appendix E for the accuracy curve on multiple hidden sizes.

| Task | U **only** | E **only** | $E_{\textbf{token}}$ **&** U **only** |
|------|------------|------------|---------------------------------------|
| Modular Addition (Train) | 47.9% | 68.3% | 100.0% |
| Modular Addition (Test) | 0.4% | 1.2% | 100.0% |
| Needle-in-a-Haystack | 17.7% | 100.0% | 98.5% |
| Decimal Addition | 27.3% | 100.0% | 48.5% |
| Parenthesis Balancing | 92.3% | 100.0% | 100.0% |

Table 2: Accuracy of embedding-only training with additional parameters fixed: optimizing only the unembedding layer, only the embedding layer, or only non-positional embeddings. Hidden sizes are all 1024; results are median over 10 random restarts. All three embedding matrices must be optimized for models to reliably complete all tasks.

trained) LSTM, a recurrent neural sequence model [22]. All models have two hidden layers, and all results are aggregated across ten random initializations.

**Random transformers learn to perform all four tasks**    Random transformers with trained embeddings and unembeddings obtain perfect accuracy on all four tasks consistently across restarts—sometimes outperforming *fully trained* recurrent networks (Table 1). These results thus point toward the role of a transformer-specific inductive bias in the effectiveness of embedding-only training.

**In general, embedding and unembedding parameters must both be trained**    To further identify which model components must be optimized to obtain these results, we consider several variants of embedding-only training: (a) leaving both token and positional embeddings fixed (so only the unembedding is trained); (b) leaving the unembedding layer fixed (so only the embedding is trained); and (c) leaving positional embeddings fixed (so token embeddings and the unembedding is trained). Results are shown in Table 2. All variants fail to reach perfect accuracy on at least one task. Notably, the variant that only trains the unembedding is unable to reach near-perfect accuracy on *any* task.

**Random transformers exhibit interpretable attention patterns**    A closer examination of the trained random models reveals similar mechanistic behaviors to their fully trained counterparts. For example, we observe attention patterns similar to "induction heads" [34] previously described in fully trained models for associative recall tasks (Figure 2).

**Random transformers use structured embeddings**    In the modular addition task, learned embeddings form circles in low-dimensional subspaces, another phenomenon observed in fully trained models for these tasks [26, 31] (Fig. 3). To better understand similarities between these models and their fully trained counterparts, we also computed the *distance irrelevance* and *gradient symmetricity* metrics described by Zhong et al. [52] for distinguishing between networks that perform modular arithmetic via the "Clock" or "Pizza" algorithm. We find a gradient symmetricity of $0.88$ and a distance irrelevance of $0.88$, consistent with a Clock-like solution.

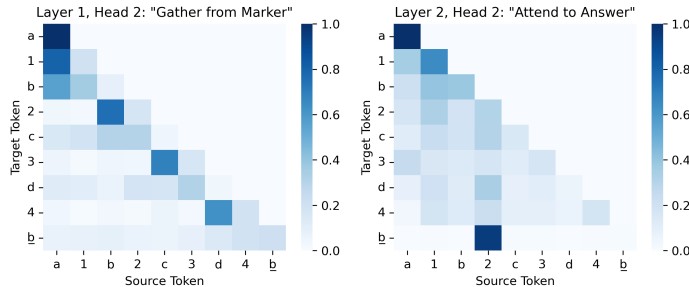
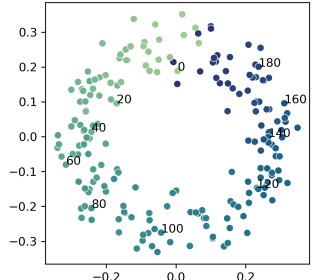

Figure 2: Attention patterns observed in a 2-layer 1024-width random transformer trained on the needle-in-a-haystack task. The input sequence is a 1 b 2 c 3 d 4 b. The layer-1 head is used by values to attend to their markers, and the layer-2 head is used by the query to attend to its associated value.

Figure 3: Circular embedding observed in random transformer on modular addition. We plot the projection of the embedding matrix onto its first and third principal components, with tokens colored according to their numeric value.

| Transformer | Accuracy | Memorized bits | Trainable parameters | Bits per parameter |
|---|---|---|---|---|
| Normal | 80.54% | 1900177 | 659584 | 2.88 |
| Random | 4.53% | 106876 | 262784 | 0.41 |

Table 3: Comparison of normal and random transformer in the memorization task.

## 5 Random Transformers Can Memorize and Generate Structured Sequences

The preceding experiments evaluated the ability of random transformers to implement single, highly structured input–output mappings. Can these models scale to more challenging tasks, involving memorization of arbitrary associations or even free-form text generation?

### 5.1 Memorization

Past work by Allen-Zhu and Li [2] has found that fully trained transformers can store roughly two bits of input per parameter. We investigate whether a similar scaling trend holds for random transformers. We study a simple memorization task in which we generate a random mapping from a two-integer key to a one-integer value, with all integers ranging from 1 to 512. Such a function requires 9 bits per input–output mapping to represent ($\log_2 512 = 9$), and may be defined for up to 262144 ($= 512^2$) values. Unlike the algorithmic tasks above, here the learned function *must* be fully specified by embeddings rather than the pre-trained model, and these experiments mainly evaluate how efficiently information can be stored in these embeddings.

We evaluate fully trained and random transformers of width 128 and 2 layers (Table 3). We measure success using an exact-match metric—an input–output pair is considered to be successfully memorized if the output token assigned highest probability by the model matches the training data. Fully trained transformers memorized 80% of training examples, stored 2.9 bits per parameter, while random transformers memorized only 5% of examples, corresponding to 0.4 bits per *trainable* parameter.

### 5.2 Language Modeling

Modeling natural language requires both memorization of arbitrary associations (e.g. between words and their parts of speech, as in Section 5.1), and structured sequence generation procedures (e.g. to enforce subject–verb agreement and close quotation marks, as in Section 4). Can random transformers make any headway on this task?

We train models on the `TinyStories` dataset, a collection of easy-to-understand stories generated by GPT-3.5 and GPT-4 [14] which have been shown to induce fluent text generation when used to train

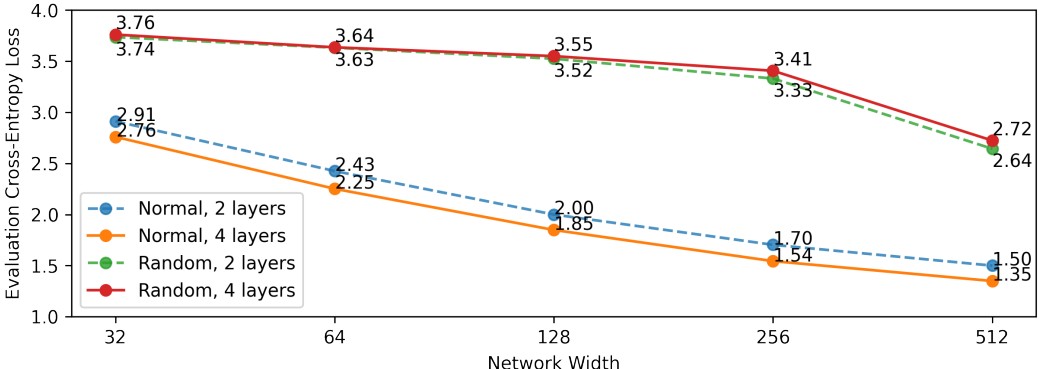

Figure 4: Language modeling performances (measured in cross-entropy loss or equivalently log perplexity, lower is better) for fully trained and random transformers. Comparatively large hidden sizes are needed for random models to match the performance of fully trained models.

---

Once upon a time, there was a little boy named Max. Max loved to play with his toy blocks. He had blocks of all colors and shapes. Max liked to build tall towers and big castles. One day, Max was playing with his blocks in his room. He built the tallest tower he had ever made. It was so tall that it almost touched the ceiling! Max was very proud of his tower. Suddenly,

**Reference (GPT-4o):** ... Suddenly, Max's cat, Whiskers, ran into the room. Whiskers saw the tall tower and thought it was a fun thing to play with. He jumped up and knocked the tower over. The blocks fell all over the floor. Max was very sad. He had worked so hard to build the tower, and now it was gone.

**Fully Trained Transformer:** ... Suddenly, Max's little sister came into the room. She wanted to play with Max's blocks too. But Max didn't want to share. He wanted to keep playing with his blocks. His sister started to cry and Max didn't understand why his sister was so upset. He thought he was just playing with them. So, he went to his room and looked for the blocks. After a while, he found them under the bed. Max was so happy that he hugged his sister and his sister. They played together and had a lot of fun. The end.

**Random Transformer (Completion 1):** ... Suddenly, Max started to blow. Max saw a loud noise coming from the tower. Max's tower was very high and strong. Max wanted to play with his blocks. Max was very proud of himself for being so helpful. Max decided to take care of his tower and his tower. Max was so happy that he had done. Max wanted to keep himself. He had to be so much fun and proud of himself. Max was happy to be able to help. Max was able to save the tower and help his tower. Max was so proud of himself. Max was happy to have such a good tower and could do it.

**Random Transformer (Completion 2):** ... Suddenly, Max's tower fell down and landed on his floor. Max fell down and hurt his knee. Max cried and came to him. Max was very sad and didn't want to get hurt. Max's feelings. Max didn't want to get away. Max just wanted to make it so badly! Max tried to take it to break. Max put him back on the floor. Max tried to make up but Max didn't know what to do. Max was wrong. Max said, "Don't worry, Max. Max are very bad and Max. Max didn't want to share it with Max. Max to play with Max and Max. Max was very sad. Max loved to play with his friends and Max played together and had lots of fun. Max played together all day. Max were happy that Max learned that sharing is caring. Max learned that being selfish and always had learned that sometimes things can share and others can make others happy again.

Figure 5: Sample completion generated by fully trained and random 512-width 2-layer transformers. While random models produce less coherent than fully trained models, they nonetheless generate text that is largely grammatical topically appropriate.

---

smaller models. As in previous sections, we evaluate both embedding-only and full training with 2- and 4-layer transformers with various hidden sizes. Scaling curves are shown in Fig. 4. Fully trained models obtain a cross-entropy loss of $1.35$, on par with the results reported by Eldan and Li [14]. Our trained random transformer with $512$ width and ~10M trainable parameters achieved a cross-entropy loss of $2.64$, roughly on par with the fully trained ones with $32$ width and only ~0.7M trainable parameters. Moreover, adding additional hidden layers does not appear to improve performance performance—2-layer random transformers in fact achieve better losses than 4-layer models.

Example outputs sampled from these models (on a newly generated prompt) are shown in Fig. 5. Even though random models obtain significantly worse perplexity than fully trained models, they could still perform several key sub-tasks needed for language generation: generated sentences are generally grammatical, topically coherent with the prompt, correctly resolve some long-distance dependencies (e.g. references to the name *Max*) and perhaps even high-level narrative structure.

## 6 Random Transformers Operate in Low-Dimensional Subspaces

Can we explain the success of random transformers in the tasks studied above? In this section, we present evidence that embedding-only training steers the hidden computation in transformers into low-dimensional subspaces in which target functions are already implemented. We term this phenomenon **subspace selection**, and show that it is distinct from *sparsification*, as these subspaces are distributed across neurons.

In Section 6.1, we measured fraction of activation variance explained by top principal components in various tasks. For algorithmic tasks, we show that both normal and random transformers work in low-dimensional subspaces, which are sufficient for solving these tasks (Appendix F). However, for language modeling and memorization, the random transformers displayed more subspace selection compared to the fully trained ones, and as a result, they attained lower performances. In Section 6.2 we constructed a task that explicitly requires operating on high-dimensional spaces, circuit imitation, and indeed, the random transformers exhibit significant performance gap compared to normal transformers.

### 6.1 Low-Dimensional Hidden Representations in Algorithmic and Memorization Tasks

To characterize the geometry of random transformers' internal representations, we present models (trained for all previously described tasks) with a set of randomly chosen inputs and collect their embeddings and hidden representations of these inputs at different layers. Using these representations, we perform two analyses: (1) the fraction of variance explained by the top **principal components** of these hidden representations (which will be large if representations lie in a low-dimensional subspace), and (2) the fraction of variance explained by the most variable entries in hidden state vectors, or **neurons** (which will be large if computation is confined to a sparse sub-network).

Both fully trained and random transformers exhibit subspace selection but not sparsification (Table 4 top) in the four algorithmic tasks. In Appendix F, we show that this behavior is expected, insofar as all four algorithmic tasks can be solved by shallow transformer-like circuits that operate in low-dimensional subspaces. On memorization and language modeling tasks, random transformers become much more concentrated on a small subspace than fully trained transformers, thus using a lower effective dimension (Table 4 bottom and Table 5). In the language modeling task, more than 30% of variance in hidden representations is explained by 10 components.

### 6.2 Subspace Selection in Circuit Imitation

To provide another window into these results, we characterize *how large* the hidden representations of a random model must be for it to simulate a random circuit that operates in a low-dimensional subspace.

To do so, we first generate a small, random *target* transformer associated with a distribution over strings $\tilde{p}$, then perform embedding-only training in a different, randomly initialized transformer to simulate its behavior on some domain of interest by minimizing:

$$\arg\min_{E,U} \ \mathbb{E}_x[\text{KL}(\tilde{p}(\cdot \mid x) \parallel p(\cdot \mid x; E, F, U))]$$

To construct target models, we begin with the same initialization scheme described in Section 3, then we scale the query and key parameters in the attention mechanism by a factor of 10, and and the feed forward weights and biases by a factor of 20. We also scale the final projection layer by $100/\sqrt{\text{width}}$. (This initialization scheme increases variability of attention patterns and target model predictions across random restarts; see Appendix D.2.2 for additional discussion.)

| Task and Transformer Type | | Principal Component Basis | | | Neuron Basis | | |
|---|---|---|---|---|---|---|---|
| Task | Model | Emb | L1 | L2 | Emb | L1 | L2 |
| **Modular Addition** | Normal | 42.0% | 21.5% | 13.8% | 3.8% | 1.6% | 3.4% |
| | Random | 72.9% | 63.6% | 44.7% | 2.8% | 3.3% | 2.9% |
| **Multi-digit Addition** | Normal | 45.3% | 87.0% | 87.3% | 1.7% | 3.3% | 2.6% |
| | Random | 55.1% | 72.7% | 63.5% | 2.0% | 3.9% | 2.8% |
| **Needle-in-a-Haystack** | Normal | 35.5% | 83.5% | 47.7% | 1.5% | 5.2% | 2.8% |
| | Random | 31.3% | 30.0% | 21.5% | 1.8% | 1.8% | 1.6% |
| **Balanced Parentheses** | Normal | 72.0% | 99.4% | 98.2% | 2.9% | 7.8% | 18.0% |
| | Random | 74.9% | 85.8% | 80.9% | 4.9% | 4.0% | 3.7% |
| **Memorization** | Normal | 15.0% | 25.3% | 23.4% | 10.2% | 12.2% | 10.5% |
| | Random | 27.5% | 27.5% | 24.7% | 9.9% | 9.8% | 9.7% |

Table 4: Median explained variance from top 10 directions under principal and neuron basis. Activations after embedding (*Emb*), layer 1 (*L1*) and layer 2 (*L2*) are collected from multiple trained 2-layer models of width 1024 and 128 (for memorization). *Normal* transformers are fully trained while *Random* transformers have only embedding and unembedding layers trained, as in previous experiments. Across tasks, a large fraction variance in models' hidden representations is explained by a small number of principal components, but these components do not appear to be aligned to individual neurons or sparse sub-networks.

| Task and Transformer Type | | Layer | | | | |
|---|---|---|---|---|---|---|
| Task | Model | Emb | L1 | L2 | L3 | L4 |
| **Language Modeling** | **Principal Component Basis** | | | | | |
| | Normal | 29.5% | 27.3% | 24.3% | 23.2% | 17.6% |
| | Random | 43.0% | 30.2% | 31.8% | 32.1% | 31.7% |
| | **Neuron Basis** | | | | | |
| | Normal | 8.6% | 16.4% | 14.7% | 13.9% | 5.9% |
| | Random | 3.3% | 2.9% | 2.8% | 2.9% | 2.9% |

Table 5: Median explained variance from top 10 directions under principal and neuron basis for language modeling task. Activations after embedding (*Emb*) and after every layer (*L1, L2, L3, L4*) are collected from trained 4-layer models of width 512. As above, a substantial fraction of variance is explained by a small number of principal components, especially in random transformers.

We evaluate fully trained and random transformers' ability to fit the target distribution for target models with a 512-token vocabulary, three layers, two attention heads, and varying hidden dimensions. Training inputs $x$ consist of 40-token sequences generated uniformly at random.

Results are shown in Fig. 6. In general, random transformers can only match the behavior of shallower (3 vs 1) and or significantly narrower (512 vs 128) models, with a sharp increase in error moving from $12 \rightarrow 16 \rightarrow 32$-dimensional models, suggesting that random transformers may *only* be able to learn computations that can be performed in lower-dimensional subspaces.

## 7 Discussion

We have shown that transformer sequence models can accomplish a variety of meaningful tasks when only their embedding layers are optimized. For tasks involving memorization, these results show that much of models' "knowledge" can be encapsulated by input embeddings rather than models' internal parameters. For more algorithmic tasks like arithmetic and parenthesis balancing, which require relatively sophisticated circuits to perform, our experiments show that versions of these circuits can be accessed in random transformers (but not LSTM sequence models) simply by constructing appropriate input and output embeddings that confine models' internal states to low-dimensional subspaces in which these tasks are performed.

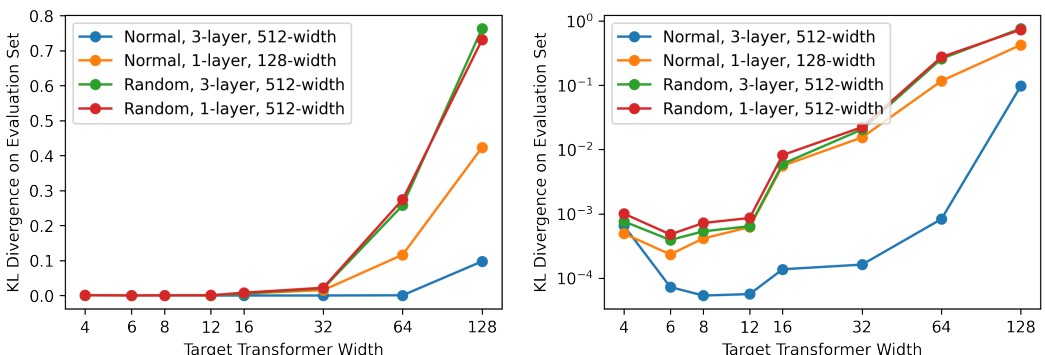

Figure 6: Kullback–Leibler divergence of circuit imitation with fully trained and random transformers (the lower the better). Both plots show the same set of results with different scales (linear and log) on the vertical axis.

However, our experiments have also highlighted several important differences between embedding-only and fully trained models, especially with regard to their parameter efficiency and information capacity. This paper leaves open the question of how computation in embedding-only models relates to fully trained ones—e.g. whether, during full training, the mechanisms we have discovered here evolve gradually into their fully trained forms, or whether fully trained models use entirely different pathways [10].

We anticipate that these random transformers will also provide an interesting new test-bed for interpretability research, and future work might investigate how learned feature codebooks [12, 6] and automated neuron labeling procedures [21, 5, 33] behave when applied to these models. Even more generally, these results motivate a closer study of the behavior of untrained models as a source of insight into differences between, and improvements upon, existing neural model architectures.

## Acknowledgement

We would like to thank MIT SuperCloud for computational resources and Mingyang Deng for valuable discussions.

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

# Supplementary material

## A    Limitations

Even within the class of transformers, the space of architectural decisions (both around model size and implementation of attention mechanisms, normalization procedures, tokenization, etc.) is very large; our experiments in this paper generally characterize a small part of this phase space. It is thus possible that some of the described trends will change as models grow or differ in parameterization details. Outside Section 4, our experiments have focused on standard transformer models, and do not answer whether these trends hold in other related linear attention [9, 35] or state-space model [19] families. Our discussion in Section 6 focused only on linear subspaces and used principal component analysis as the primary tool, but it is also possible that subspaces or latent semantics appear non-linearly which is not addressed by our current analysis.

## B    Impact statement

We do not anticipate any ethical concerns associated with these results and we believe understanding AI systems is a crucial step in harnessing their power for good.

## C    Computational Resources

Roughly 154 GPU days of NVidia V100 were spent on this project.

## D    Setup details

### D.1    Data Curation and Tokenization Details

#### D.1.1    Modular Addition

Fix modulus $p = 199$. We randomly shuffle all possible inputs ($p^2$ of them) perform a $95\%$: $5\%$ for training and test set. The split is the same (generated with the same seed and procedure) for all runs across all architectures.

#### D.1.2    Dynamically Generated Tasks

For these tasks, we used a stream of training data and a heldout test set. The test set is fixed for all runs across all architectures. The training data is generated on the fly within the same distribution to avoid overfitting.

**Needle-in-a-Haystack**    The number of entities (marker-value pairs) is first uniformly generated in $[1, 30]$. The values are generated as integers in $[1, 127]$. The markers are generated as distinct integers in $[127 + 1, 128 + 30]$. The final query token is the asked marker (uniformly chosen) plus 30. For example, a 1 b 2 c 3 d 4 b̲ in token ids would be $[128, 1, 129, 2, 130, 3, 131, 4, 159]$ and the expected output will be token 2.

**Decimal Addition**    Fix the number of digits $l = 10$. The two numbers to be added are independently uniformly generated within $[10^9, 10^{10} - 1]$. The added numbers and the results are reversed. The addition sign has token id 10 and the equal sign has id 11. For example, $1111111112 + 2222222223 =$ in token ids would be $[2, 1, \cdots, 1, 10, 3, 2, \cdots, 2, 11]$. The result uses $[20, 29]$ to represent digits and 30 to signal the end of output. For the example, the expected output will be 3333333335 encoded as $[25, 23, \cdots, 23, 30]$.

**Parentheses Balancing**

The opening parenthesis has token id 1, the closing parenthesis has token id 2, and the question mark has token id 3. For the result, 2 signals balanced and 1 signals unbalanced. For example, (())()? in token ids will be $[1, 1, 2, 2, 1, 2, 3]$ and the expected output is balanced, token 2.

As sequences of uniformly random parentheses are (a) likely unbalanced (b) easier to validate, we performed a generate-then-mutate strategy to generate strong data for the task.

*Generate* With $1/3$ probability, we generate a random sequence of at most $60$ parentheses (length uniformly chosen, then independently all the parentheses). With $2/3$ probability, we generate a balanced parentheses sequence. We first uniformly choose the number of pairs of parentheses $t$ in $[1, 30]$. We then generate recursively: when $t \geq 2$, with $1/2$ probability we generate a $(t-1)$-parentheses sequence recursively and add a pair of parentheses on the outside, with $1/2$ probability we uniformly sample $u \in [1, t-1]$ and output the concatenation of a $u$-parentheses and a $(t-u)$-parentheses sequence generated recursively.

*Mutate* With $1/2$ probability, we choose some random index pairs and swap the corresponding parentheses. Independently with $1/2$ probability, we (then) choose some random indices and flip the corresponding parentheses. The number of pairs and indices are sampled according to a geometric distribution.

**Circuit Imitation** The inputs are uniformly random $40$-token sequences of integer tokens $[0, 511]$. The target output is the distribution from a target transformer generated as in Appendix D.2.2 below. The width of the target transformer is chosen in $[4, 128]$ (Figure 6).

### D.1.3 Memorization

For every integer pair $x \in [0, 511]$, $y \in [512, 512 + 511]$, we generate one data point with input $[x, y]$ and output $z$ uniform in $[0, 511]$. There is no "test split" as the goal is to memorize all the associations.

### D.1.4 Language Modeling

We used the original train-test split of the TinyStories dataset [14]. We trained a 10000-token BPE tokenizer from the training split alone.

## D.2 Model Details

### D.2.1 Transformers

We use the GPT-2 [38] implementation of Huggingface [48]. Dropout and weight tying are disabled. The activation function is kept as the default GeLU [20].

Specifically, all the weights of feed-forward layers are initialized by sampling from isotropic Gaussians with mean $0$ and standard deviation $0.02/\sqrt{2n}$ where $n$ in the number of layers. All the bias matrices are initialized with zeroes. All the other weight matrices (including key, value, query, embedding, unembedding matrices) are initialized with $0$-mean Gaussians with standard deviation $0.02$. The affine transformations in layer normalizations are initialized as identity.

We used two layer transformers except for the language modeling and circuit imitation task. The number of parameters for the algorithmic tasks can be found in Table 6.

| Task | $E_{\text{token}}$ | $E_{\text{pos}}$ | $U$ | **Intermediate Layers** $F$ |
|---|---|---|---|---|
| Modular Addition | 99,328 | 5,120 | 99,328 | 25,194,496 |
| Needle-in-a-Haystack | 262,144 | 102,400 | 262,144 | 25,194,496 |
| Decimal Addition | 31,744 | 40,960 | 31,744 | 25,194,496 |
| Parenthesis Balancing | 4,096 | 81,920 | 4,096 | 25,194,496 |

Table 6: Number of parameters of each type for the 2-layer 1024-width transformers in Section 4.

### D.2.2 Target Transformer in Circuit Imitation

Based on the initialization in Appendix D.2.1, we make the following modifications.

**Feed-forward Layers:** Standard deviation scaled up by 20x: changed to $0.4/\sqrt{2n}$.

| Task and Transformer Type | | Principal Component Basis | | | Neuron Basis | | |
|---|---|---|---|---|---|---|---|
| Task | Model | Emb | L1 | L2 | Emb | L1 | L2 |
| Decimal Addition | Normal | 40.7% | 91.6% | 89.6% | 3.3% | 6.4% | 6.8% |
| | Random | 73.3% | 62.6% | 51.4% | 4.3% | 4.0% | 3.6% |
| Needle-in-a-Haystack | Normal | 31.4% | 75.4% | 42.5% | 2.8% | 5.6% | 3.9% |
| | Random | 51.3% | 42.6% | 33.9% | 4.4% | 4.6% | 3.9% |
| Balanced Parentheses | Normal | 47.5% | 100.0% | 99.9% | 7.0% | 11.1% | 9.4% |
| | Random | 85.2% | 89.1% | 84.7% | 9.7% | 8.0% | 6.7% |

Table 7: Median explained variance from top 10 directions under principal and neuron basis collected from width 512 transformers. Rounded to one decimal piece. See Table 4 for more details.

**Attention Matrices (key, value, query):** Standard deviation scaled up by 10x: changed to $0.4$.

**Unembedding Matrix:** Standard deviation changed to $2/\sqrt{n}$.

After such modifications, we measured the entropy of output distribution on two random inputs. Across all the widths, the mean entropy stays within $[2.89, 3.02]$. We also measured similarity of output distributions on different inputs to prevent mode-collapse-like results, and the mean KL divergence of output distributions on random inputs is $[0.8, 3.3]$ across all widths.

### D.2.3   LSTM

We used textbook long short-term memory [22] with an encoding layer and an unembedding (projection) layer added.

### D.3   Training Details

For synthetic experiments, we used AdamW optimizer [27] with a learning rate $10^{-3}$ and weight decay $10^{-3}$. For LSTM a learning rate $5 \times 10^{-3}$ is used for faster convergence. For the language modeling task, we used AdamW optimizer with a learning rate $6 \times 10^{-4}$ and weight decay $0.1$. We clip all gradient norms at $1$.

For random transformer experiments, the intermediate (query, embedding, unembedding, feed-forward) layers are kept as randomly initialized. We use a fixed number of training steps and no early stopping.

**Modular Addition:** 5000 epoches. Batch size 4000.

**Needle-in-a-Haystack, Decimal Addition, Parentheses Balancing, Circuit Imitation:** $10^4$ steps of batch size 1000. Again, the training data is generated dynamically.

**Memorization:** 21000 epoches. Batch size $2^{15}$.

**Language Modeling:** 5 epoches. Batch size 20 and context window 512.

## E   Performance on synthetic tasks across model scales

In Fig. 7 we provide the test accuracy of needle-in-a-haystack, decimal addition and parenthese balancing for fully trained and random transformers across different model widths. Note that the 1024-width fully trained transformer had trouble reaching perfect accuracy in parentheses balancing, likely due to imperfect hyperparameter choices.

We also include the explained variance measurements on 512-width models in these three tasks for completeness (Table 7). Generally more variances are explained from the top directions as the models are narrower.

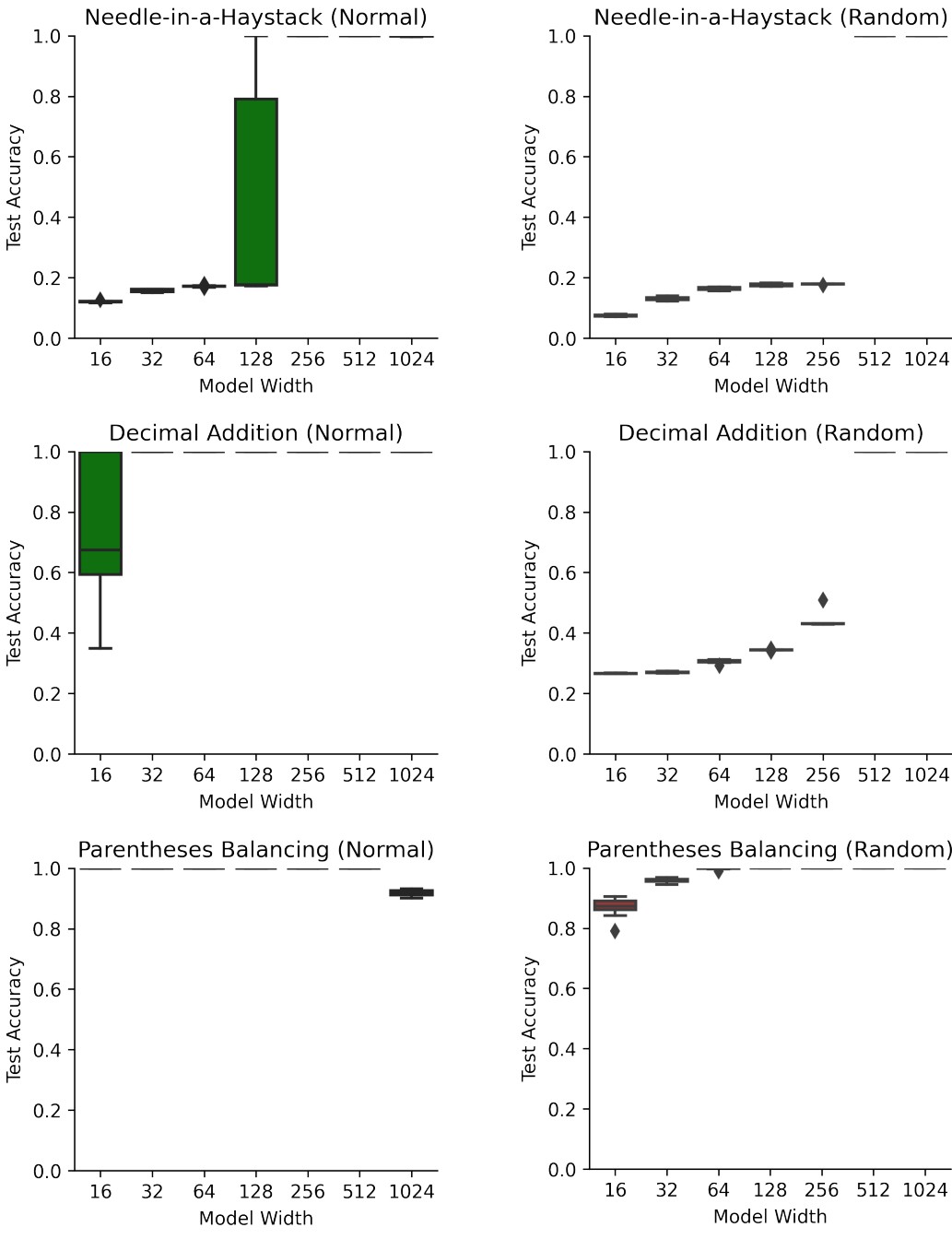

Figure 7: Box plot of test accuracy of synthetic tasks across model scales.

## F  Constant-dimensional subspaces are sufficient for many synthetic tasks

**Definition F.1** (Intermediate complexity of tasks (informal))**.** For a family of neural networks with varying intermediate width, we define the **intermediate complexity** of a task as the minimal width of *intermediate* layers required to succeed in the task.

For example, a 2-layer transformer with intermediate width 2 starts by embed the tokens into the 2-dimensional hidden space, passes them through two transformer blocks of width 2, then linearly

project the activation for the final output (unembedding). If one task could be solved with one such transformer, its intermediate complexity for 2-layer transformers would be at most 2.

If we consider the activations as the working memory of the LLM, intuitively a task has low intermediate complexity if it requires a low working memory per token. In the following, we show that all the synthetic tasks we proposed indeed have constant intermediate complexity for 2-layer transformers, thus they only require a constant-dimensional intermediate subspaces to work on.

**Modular Addition** We may use a transformer to implement three instances of the *Pizza* algorithm described in Zhong et al. [52] (multiple copies are needed to avoid the "antipodal pairs" problem described there). Each implementation of this algorithm requires intermediate representations of size 2, so the intermediate complexity for 2-layer transformers is $\leq 6$.

**Needle-in-a-Haystack** We may use a transformer to implement induction heads [34] as follows: We first store each token's preceding token in its activations by attending to the previous position. We then retrieve the answer by attending to the position storing the matching token in the previous step. Both steps could be done by using attention to compute squares of differences and take the minimum. The number of hidden states needed to distinguish each symbol scales only with the size of the vocabulary, and is at most equal to the vocabulary size, so the intermediate complexity is at most twice the vobaulary size.

**Parentheses Balancing** Let the given sequence of parentheses be $p_1, p_2, \cdots, p_n$. Let $x_i = \sum_{j=1}^{i} c_i$ where $c_i = 1$ if $p_i$ is an open parenthesis and $c_i = -1$ if $p_i$ is a closing parenthesis. The given sequence is a valid parentheses sequence if and only if $x_n = 0$ and $\min_{i=1}^{n} x_i = 0$. In a transformer, we can first compute $x_i/i$ with a uniform attention. Then, we attend to the position with minimum $x_i/i$, breaking ties by position (let the attention on $i$ pre-softmax be $T(-x_i/i + \epsilon \cdot i)$ for large enough $T$ and small enough $\epsilon > 0$). For valid parentheses sequences, the last position should only attend to itself. We then check if the attended position is indeed the last one and has $x_i/i = 0$ by computing squares of differences. The intermediate complexity for 2-layer transformers is thereby again constant. (note that a different construction for bounded-depth Dyck was given in [50])

**Decimal Addition**

Let the reversed list of digits of two numbers and their sum be $a_1, a_2, \cdots, a_n$, $b_1, b_2, \cdots, b_n$, and $c_1, c_2, \cdots$. Let $x_i = \sum_{j=1}^{i} a_j 10^{j-i}$, $y_i = \sum_{j=1}^{i} b_j 10^{j-i}$, $z_i = \sum_{j=1}^{i-1} c_j 10^{j-i}$, we have $c_i = (x_i + y_i - z_i) \bmod 10$: $a + b \equiv \sum_{j=1}^{i} 10^{j-1}(a_j + b_j) \pmod{10^i}$, $(a+b) \bmod 10^{i-1} = \sum_{j=1}^{i-1} c_j 10^{j-1}$, and $c_i = ((a+b) \bmod 10^i - (a+b) \bmod 10^{i-1})/10^{i-1}$.

**Prepare:** Take small $\epsilon > 0$. For positions $1, 2, \cdots, n$, let the positional embedding of position $i$ be $[1, 10^i, 10^{-i}\epsilon]$. In the first head of the first layer, take the first dimension in the positional embedding for both Q and K, so we get a uniform attention on the prefix from which $\alpha_i = \frac{1}{i} \sum_{j=1}^{i} a_j$ can be calculated. In the second head of the first layer, take the third dimension in the positional embedding for Q and the second dimension in the positional embedding for K, so the contribution from the $j$-th position to the $i$-th position is $\frac{1}{Z_i} e^{\epsilon 10^{j-i}} \approx \epsilon \frac{1}{Z_i}(10^{j-i} + 1)$ ($e^c \approx 1 + c$ for $0 < c \ll 1$) for normalizing constant $Z_i$, we can then calculate $\beta_i = \epsilon \frac{1}{Z_i} \sum_{j=1}^{i} a_j(10^{j-i} + 1)$. We then have $x_i = \frac{Z_i}{\epsilon} \beta_i - i\alpha_i$ which we will utilize in the next step. Similarly, we can have $y_i$ and $z_i$ ready in the corresponding position (position of $b_i$ and $c_{i-1}$).

**Generate** $x_i + y_i - z_i$**:** In the second layer, attend from the position of $a_i$ and $b_i$ at the position of $c_{i-1}$. Specifically, set the attention from the $a_k$'s position to the $c_{i-1}$'s position be $\epsilon i - \Lambda \cos((k - i)/n) = \epsilon i - \Lambda(\cos(k/n)\cos(i/n) + \sin(k/n)\sin(i/n))$ pre-softmax. This could be done using three dimensions in the positional embeddings. We also set the attention from the plus sign to the $c_{i-1}$'s position be 1 pre-softmax. The attention from $a_k$ to $c_{i-1}$ will be negligible if $k \neq i$ and will be proportional to $\epsilon i$ post-softmax for $k = i$. We can then calculate $-i\alpha_i$ from it and similarly $\frac{Z_i}{\epsilon} \beta_i$, and thus $x_i$, $y_i$ and $z_i$.

**Approximate the Answer:** As $x_i + y_i - z_i$ is an integer from $[0, 19]$ and we can also compute an affine transform of it, we may now simply proceed with the universal approximation theorem. As an

example, in ReLU networks we may check if $x_i + y_i - z_i = v$ by noticing

$$\text{ReLU}(x_i+y_i-z_i-(v+1))+\text{ReLU}(x_i+y_i-z_i-(v-1))-2\text{ReLU}(x_i+y_i-z_i-v) = [x_i+y_i-z_i = v],$$

So to create an indicator variable $[x_i + y_i - z_i = 1]$ we simply need to generate $x_i + y_i - z_i$, $x_i + y_i - z_i - 1$, $x_i + y_i - z_i - 2$, pre-ReLU.

The intermediate complexity for 2-layer transformers is thereby again constant. This approach is quite crude so it is likely that the constant can be greatly improved.

## G   Difficulty of the synthetic tasks

Chomsky hierarchy has been found to be predictive of the neural architectures' performances, and especially length-generalization performances, in algorithmic setups [13]. We list the Chomsky hierarchy of the synthetic tasks we picked in Table 8.

| Task | Chomsky Hierarchy |
|---|---|
| Needle-in-a-Haystack | Regular |
| Decimal Addition | Context Sensitive |
| Parenthesis Balancing | Context Free |

Table 8: Classification of tasks according to the Chomsky hierarchy. For the decimal addition task, we consider the most flexible setting where all the strings representing decimal additions (does not have to be equal-lengthed) are considered within the language.

We also examined a baseline with linearized transformer. In this variant of transformer, the attention matrix ($\text{softmax}(QK^T)$) is replaced with a lower diagonal matrix of 1s. In other words, the attention is replaced by a simple token prefix sum mechanism. We tested such transformers with width 512 and 2 layers on the synthetic tasks as a baseline performance. The result is shown in Table 9. We can see that such transformers have large performance gaps with normal transformers, confirming the difficulty of our chosen tasks.

| Task | Accuracy (%) |
|---|---|
| Needle-in-a-haystack | 17.67 |
| Decimal Addition | 26.39 |
| Parenthesis Balancing | 97.32 |

Table 9: Linearized transformer performance in terms of accuracy.

## H   Accuracy curve of the memorization task

The accuracy curve in the memorization task during training is shown in Fig. 8. Training of both transformers has converged.

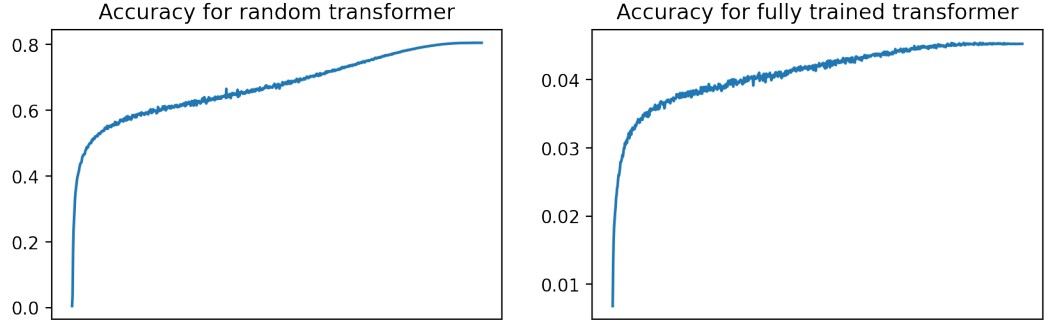

Figure 8: During training, the accuracy curve from fully trained and random transformers in the memorization task (Section 5.1). Note that the evaluation set is exactly the training set as the goal is to memorize.

