# OpenReview forum: "Algorithmic Capabilities of Random Transformers"
_NeurIPS.cc/2024/Conference — NeurIPS 2024 poster_

### Official Review · Reviewer_LwAU · 2024-07-06

**Soundness:** 3
**Presentation:** 1
**Contribution:** 2
**Rating:** 4
**Confidence:** 3

**Summary:**

This paper explores the inherent algorithmic capabilities of randomly initialized transformer models, particularly focusing on the functions that can be learned when only the embedding layers are optimized. It demonstrates that even without training the internal transformer layers, these models can perform complex tasks such as modular arithmetic, decimal addition, and associative recall. This challenges the traditional belief that deep training is essential for achieving proficiency in these tasks.

**Strengths:**

1. **Insight into Model Initialization:** The research provides novel insights into the importance of model initialization, revealing that transformers possess intrinsic algorithmic abilities even prior to training.
2. **Interpretability and Simplicity:** The study shows that these algorithmic tasks can be accomplished with straightforward modifications to the input and output embedding layers, thereby enhancing the interpretability of transformers at the initialization stage.

**Weaknesses:**

1. **Need for Improved Writing:** The paper's writing style and organization need enhancement. For instance, there is a missing reference in line 19 of the introduction's first paragraph. Additionally, a footnote on page 4 is left empty, and another reference is missing on page 18, line 559.
2. **Generalization Concerns:** The findings are primarily demonstrated on synthetic tasks. The paper lacks a thorough discussion on the applicability of these findings to real-world datasets or tasks.

**Questions:**

In Section 7, the paper discusses storing knowledge within the token embeddings.  [1] suggests that early-site MLPs retain knowledge about the tokens. I am curious whether similar results could be achieved by randomizing the embeddings while optimizing these early-site MLPs.

[1] Meng, Kevin, et al. "Locating and editing factual associations in GPT." Advances in Neural Information Processing Systems 35 (2022): 17359-17372.

**Limitations:**

The concept presented is intriguing; however, the paper's writing quality needs improvement to meet professional academic standards.

---

> ### Author Rebuttal · Authors · 2024-08-06
>
> Thanks for the insightful review! The following are our responses.
>
> > **Need for Improved Writing:** The paper's writing style and organization need enhancement. For instance, there is a missing reference in line 19 of the introduction's first paragraph. Additionally, a footnote on page 4 is left empty, and another reference is missing on page 18, line 559.
> >
>
> We are sorry for these technical issues. We have fixed them and did some more passes on the writings. We also refined the presentation and arguments following suggestions from all the reviewers. If you have any questions with the content of the paper, please let us know.
>
> > **Generalization Concerns:** The findings are primarily demonstrated on synthetic tasks. The paper lacks a thorough discussion on the applicability of these findings to real-world datasets or tasks.
> >
>
> While our discussion is primarily focused on synthetic tasks as they correspond to well-defined abilities, we believe our work offers interesting and crucial insights that will also help us understand (fully trained) transformers better. For example, an interesting follow up question would be what abilities are newly developed in training rather than merely unlocked in a similar sense. We also examined language modeling as an example of more complicated and less well-defined task.
>
> > In Section 7, the paper discusses storing knowledge within the token embeddings. [1] suggests that early-site MLPs retain knowledge about the tokens. I am curious whether similar results could be achieved by randomizing the embeddings while optimizing these early-site MLPs.
> >
>
> This question is unrelated to our current setup as the focus of our study is the expressiveness of the network with random / untrained intermediate layers instead of embedding layers. In fact, we believe the setup where the embeddings instead of intermediate layers are left untrained is much simpler, at least in the high-width regime, as the random embedding matrix will have a high probability possessing a full row rank, from which any embedding matrix could be simulated with the correct linear matrix applied on it. More formally, say the random embedding matrix is fixed to be $E$, when the width is high enough, with high probability for any desired embedding matrix $E’$ we will have some matrix $A$ so that $EA=E’$, hence the embedding and the first linear layer combined could act as $E’$.

---

> > ### Comment · Reviewer_LwAU · 2024-08-12
> >
> > I appreciate the authors response, which addressed all my questions. I increase my score.

---

### Official Review · Reviewer_ZcHJ · 2024-07-11

**Soundness:** 3
**Presentation:** 3
**Contribution:** 2
**Rating:** 5
**Confidence:** 4

**Summary:**

The aim of the paper is to understand how much of the effectiveness of Transformer models depends on the architecture itself rather than the possibility to train its internal parameters. To this aim, the authors study causal Transformers where only the embedding matrix, the positional encodings and the final projection matrix are trained, while the rest of the parameters do not change after random initialization. The authors compare the performance of small-scale random Transformers to that of fully-trained ones (of the same size) on some synthetic and language modeling tasks. They observe that random Transformers (with properly optimized embeddings) are able to solve all the synthetic tasks, in some cases exhibiting computational circuits similar to those observed in trained models. Although random Transformers have weaker memorization capabilities, they can also generate grammatically correct (and often consistent) text.

**Strengths:**

I think that this work is original and relevant. The authors explore an interesting question drawing from the rich literature on randomly initialized neural networks and proposing a connection with circuit-based interpretability techniques. The experiments and findings are generally coherent with the conclusions drawn by the authors. The paper is well-written and the results are generally presented in a clear way.

**Weaknesses:**

In my opinion, the most critical issue is related to the use of the term “algorithmic capabilities”. In the literature this term is often misused (or even abused): a system has algorithmic capabilities when it can robustly (if not perfectly) extrapolate symbolic knowledge outside its training distribution. The synthetic problems investigated in the present study do not assess algorithmic capabilities, because the models are tested only within the ranges encountered during training. The case of modular addition is particularly representative: the authors only consider a subspace of the problem, shuffling the patterns and using 95% of them for training and 5% for testing. This is probably why performance is at ceiling even with random architectures. Comparing models using tasks that can be solved with 100% of accuracy (see Table 1) is not particularly meaningful. In light of this, I think that the significance and impact of the present work might be overall quite limited.

The results related to the attention patterns on the needle-in-a-haystack task are quite expected, since this problem can be solved by using only input embeddings (indeed, the model where the positional encodings are not trained still achieves almost perfect accuracy).

The authors should improve the presentation of the results related to the low-dimensional sub-space analysis. Saying that there exists a (small) width of hidden representations that is sufficient for a fixed-depth transformer to solve a task (Appendix F) seems analogous to saying that in a higher-dimensional space, it is enough to use a smaller number of dimensions (principal components) to solve the task. The authors should better clarify what they mean by “concentration in a low-dimensional sub-space” (even formally, if necessary) and better relate this to the notion of sparseness. The presentation of the results in section 6.2 is also not very clear: the authors say that they fit the distribution of outputs generated by a randomly initialized transformer with 3 layers and hidden representations of varying dimensions. However, they then say (and show in Fig. 6) that a random transformer can match the behavior of a shallower model of depth 1, which was not mentioned before. Finally, they comment on the capability of random Transformers of matching the behavior of significantly narrower (128) models, but the difference in linear scale (left panel in Fig. 6) appears still marked even compared to 128-width models.

**Questions:**

-	Algorithmic capabilities must be evaluated in the out-of-distribution regimen. I am not asking the authors to demonstrate that random Transformers can solve algorithmic problems OOD (this would be an extraordinary achievement), but rather to compare them with trained models using more sensitive versions of the synthetic tasks, and discuss the findings in a less overhyped fashion (starting from the title).
-	“Transformers seem to be especially well-suited at problems involving numerical reasoning”. This sentence relfects the overhyped interpretation of Transformers capabilities in algorithmic / numerical tasks (see https://arxiv.org/abs/2305.18654 and https://www.mdpi.com/2076-3417/14/2/744 for different perspectives).
-	The analyses on the low-dimensional sub-spaces should be presented in a more formal and clear way.
-	The presentation of results of a Normal 16-dimensional Transformer in Table 1 creates confusion: since similar accuracies are obtained with a Random 16-dimensional transformer (Appendix E) it would be better to include results for both Random and Normal 16-dimensional models or just show the 1024-dimensional models.
-	It would be interesting to investigate the exceptionally poor performance in modular addition (Test) of the models in which only E or U are trained (Table 2).
-	Some error bars in Fig. 7 seem too large or even misaligned. Is that a formatting problem or does it indicate poor convergence of training?
-	In the appendix, the input for the memorization task is described as two integers, one in [0, 511] and the other in [512, 512+511], which is not coherent with the description in the main text (section 5.1). Please clarify.
-	The metric described in the caption of Fig. 4 (log perplexity) is not coherent with the y-axis label (Cross-Entropy Loss). Please clarify.

**Limitations:**

The authors properly discuss the possible limitations of their work.

---

> ### Author Rebuttal · Authors · 2024-08-06
>
> Thanks for the detailed and insightful review! The following are our responses.
>
> > Misuse of term "algorithmic capabilities"
>
> Let us humbly disagree. While a good algorithm should work for all possible data, in practice *implementations* often cannot extrapolate well, which does not disprove that the model class possesses algorithmic capabilities. For example, an algorithm in C might fail with a too large input array. Similarly, if a transformer uses absolute positional embedding, it might fail outside the training range. However, we can still claim that our C program or transformer possesses algorithmic capability. We will refine our wording to clarify that the algorithmic capabilities are for the model class, not a specific trained model. We are more than happy to discuss further on this topic!
>
> > Clarify meaning of “concentration in a low-dimensional sub-space”
>
> Yes. What you stated is exactly what we are demonstrating in Sec 6.1. We have refined the introduction of Sec 6 and added a short formulation to clarify this.
>
> > Authors say and show that a random transformer can match behavior of a shallower model of depth 1, which was not mentioned before. They comment on capability of random Transformers matching behavior of significantly narrower (128) models.
>
> This is not what we meant. We are matching the behavior of a target (carefully generated) transformer *with* either random or fully trained transformers, including a shallower model of depth 1. The target transformer is well-approximated by a depth 1 transformer in the low-width regime.
>
> We agree that random transformers cannot match significantly narrower models - this is exactly we try to demonstrate here. The argument is that the random transformers can approximate or *act as* "narrow" circuits, but fail for moderately wide (128) circuits. This is their fundamental limitation. We apologize for the possible confusion and have revised this section for clarity.
>
> > Algorithmic capabilities must be evaluated OOD; Use more sensitive versions of tasks
>
> Demonstrating OOD problem-solving is out of the scope of this paper, but again we do not think this is a fundamental issue, and we want to point out that we are already using some of the more sensitive versions of this task. For the needle searching task, one common practice [1] is to insert an out-of-place sentence into a long text, easily spotted as an unconditional retrieval task. Our version requires key-value retrieval. For the parenthesis balancing task, many previous works focused on limited depth versions [2, 3], while our version features high-depth and close-to-correct data (see Appendix D.1.2).
>
> [1] Anthropic. Long context prompting for Claude 2.1, 2023. URL https://www.anthropic.com/news/claude-2-1-prompting
>
> [2] Wen et al. Transformers are uninterpretable with myopic methods: a case study with bounded dyck grammars.
>
> [3] Shunyu et al. Self-attention networks can process bounded hierarchical languages.
>
> > Transformers seem especially well-suited at problems involving numerical reasoning: overhyped
>
> Thanks for the pointers. The main point we are trying to get through here is that transformers are superior in these tasks compared to other neural architectures, which is shown in e.g. [4] and perhaps in Table 1 of our paper. We have toned down the sentence in the working version.
>
> [4] Saxton et al. Analysing mathematical reasoning abilities of neural models.
>
> > Analyses on low-dimensional subspaces should be more formal and clear
>
> We agree that our argument is somewhat convoluted and here is it rephrased. For simple synthetic tasks, working in low-dimensional subspaces suffices (see App F), so both normal and random transformers display subspace concentration. For language modeling and memorization, random transformers show more subspace concentration and thus lower performance. For a task requiring high-dimensional operations like circuit imitation, random transformers fall short. We have refined this argument and included a formal definition.
>
> > Table 1 is confusing
>
> Great suggestion! We have included the results for the random 16-dimensional transformer in Table 1 in the working copy and rebuttal pdf.
>
> > Poor performances in modular addition of models where E or U are trained
>
> An exact proof is a bit far-fetched, but both algorithms discussed in [5] requires the use of both embedding and unembedding, so if the model succeeded with only E or U trained, this suggests that a new algorithm is yet to be discovered.
>
> [5] Zhong et al. The clock and the pizza: Two stories in mechanistic explanation of neural networks.
>
> > Error bars in Fig. 7
>
> It is not a formatting issue. For example, in the 16-width normal decimal addition training, 4 out of 10 runs had perfect accuracy, while the others had no more than 67.5%, with one run at 34.9%. This discontinuity might suggest a sharp phase transition similar to grokking [6]. We have refined the plot to be a box plot (see rebuttal pdf).
>
> [6] Power, Alethea, et al. "Grokking: Generalization beyond overfitting on small algorithmic datasets."
>
> > Clarify input for memorization
>
> We did not detail the exact formatting in the main text for the sake of simplicity, but a random function with key $[0,511]\times[512,512+511]$ has the same distribution as a random function with key $[1,512]^2$. Alternatively, it could be considered as tokenizing the first input from 0 to 511 and the second from 512 to 1023. We did not ablate on this but the spirit here is to let the network focus on memorizing instead of distinguishing between the two parts of the key.
>
> > Metric log perplexity is not coherent with label Cross-Entropy Loss
>
> For a generative process $p$, its perplexity on a sequence $x_1,x_2,\cdots,x_n$ is $\exp(-\frac{1}{n}\sum_{i=1}^n \log p_\theta(x_i\mid x_{<i}))$. The cross-entropy loss on token $x_i$ is $-\log p_\theta(x_i\mid x_{<i})$, so the log perplexity equals the cross-entropy loss averaged over the sequence. We have refined the caption.

---

> > ### Comment · Reviewer_ZcHJ · 2024-08-08
> >
> > I appreciate the Authors' willingness to address the issues raised in my review and their replies to my comments. Having read the Author's responses and the comments posted by the other reviews, I opted to maintain my overall score: I still think this is a borderline paper, presenting some interesting and original ideas but (in my opinion) not always with the proper experimental support. Regarding the use of the term "algorithmic", I can see the Author's point but I believe that using it in the title and as a main argument could be misleading, since the paper does not contain experiments demonstrating a non-trivial degree of OOD extrapolation.

---

### Official Review · Reviewer_wwwU · 2024-07-11

**Soundness:** 2
**Presentation:** 3
**Contribution:** 2
**Rating:** 5
**Confidence:** 3

**Summary:**

This paper studies transformers with freezed random intermediate layers, and embedding-only trainable layers. The authors show that wide enough random transformers are capable of performing simple algorithmic tasks such as addition and parenthesis balancing. This study further investigates the reason behind such learnability, concluding that such transformers operate in low-dimensional subspaces.

**Strengths:**

* The observation that training only the embedding layers can lead to noticeable accuracy on tasks is interesting.
* The paper is well-written and easy to follow.

**Weaknesses:**

* It is not clear how this observation about random transformers is helpful and useful. Especially, given that the (1) studied tasks are very simple (2) random transformer needs to be very wide to compete with a normal transformer of much smaller width.
* In Section 6 (Random Transformers Operate in Low-Dimensional Subspaces), the conclusions are a bit mixed and confusing. The text suggest that “Random” transformers operate in low-dim subspaces, but Table 4 does not show any difference between principal components of either normal or random transformers — in some tasks the former is larger, in other tasks the later. Perhaps, we could conclude “Transformers generally operate in in low-dim subspaces on these tasks”, but this conclusion is irrelevant to the motivation of the section to investigate the success of random transformers.
* Some table results seem inconsistent: In Table 1, random transformer achieves 100% accuracy for the Decimal Addition. However, in Table 2,  “E_token & U only” achieves only 48.5%. From my understanding, these two numbers should match, unless they are run under different settings. The “Needle-in-a-Haystack” tasks seems to have the same issue as well. Please correct me if I am missing something here.

**Questions:**

* From my understanding of the attached code, the LSTM is trained without gradient clipping. Since LSTMs are generally harder to train, and require small learning rates, with gradient clipping and larger number of optimization steps, I wonder if their performance would match with a normal Transformer in Table 1 if we apply such improvements? In other words, is it an optimization issue that LSTM is lagging behind, or a generalization issue?

**Limitations:**

Yes

---

> ### Author Rebuttal · Authors · 2024-08-06
>
> Thanks for the detailed and insightful review! The following are our responses.
>
> > It is not clear how this observation about random transformers is helpful and useful. Especially, given that the (1) studied tasks are very simple (2) random transformer needs to be very wide to compete with a normal transformer of much smaller width.
> >
>
> One main takeaway of the work, we believe, is that circuits capable of solving these simple tasks naturally exist in randomly initialized transformers, and could be activated merely by directing to these circuits by tuning the embedding and unembedding layers. We are not trying to advocate for actually employing them in everyday tasks instead of fully trained transformers, but we believe our work offers interesting and crucial insights that will help us understand (fully trained) transformers better. For example, an interesting follow-up question would be what abilities are newly developed in training rather than merely activated in a similar sense. We also studied language modeling as an example of a more complicated and less well-defined task.
>
> > In Section 6 (Random Transformers Operate in Low-Dimensional Subspaces), the conclusions are a bit mixed and confusing. The text suggest that “Random” transformers operate in low-dim subspaces, but Table 4 does not show any difference between principal components of either normal or random transformers — in some tasks the former is larger, in other tasks the later. Perhaps, we could conclude “Transformers generally operate in in low-dim subspaces on these tasks”, but this conclusion is irrelevant to the motivation of the section to investigate the success of random transformers.
> >
>
> This is a good point and the following is our argument, rephrased. For the four simple synthetic tasks, it is known that working in low-dimensional subspaces suffices (see App F) so both normal and random transformers display some degree of subspace concentration (and indeed, they can all reach perfect or near-perfect accuracy in these tasks). For language modeling and memorization, the random transformers display more subspace concentration compared to the fully trained ones and as a result, have lower performances. Finally, we show that for a task that explicitly requires operating on high-dimensional spaces, circuit imitation, the random transformers fell short compared to normal transformers. Agreed that this argument is not clearly expressed in the paper and we are working to refine that.
>
> > Some table results seem inconsistent: In Table 1, random transformer achieves 100% accuracy for the Decimal Addition. However, in Table 2, “E_token & U only” achieves only 48.5%. From my understanding, these two numbers should match, unless they are run under different settings. The “Needle-in-a-Haystack” tasks seems to have the same issue as well. Please correct me if I am missing something here.
> >
>
> We agree it is somewhat confusing. In the paper, by embedding we mean both the token embeddings E_token and the positional embeddings E_pos. It is the positional embeddings that are left untrained / as randomly initialized here, which accounts for the lowered performances. We also noted this in the table caption.
>
> > From my understanding of the attached code, the LSTM is trained without gradient clipping. Since LSTMs are generally harder to train, and require small learning rates, with gradient clipping and larger number of optimization steps, I wonder if their performance would match with a normal Transformer in Table 1 if we apply such improvements? In other words, is it an optimization issue that LSTM is lagging behind, or a generalization issue?
> >
>
> We in fact clipped the gradient norm to 1, which is the default in the huggingface trainer, and hand-tuned the learning rate of LSTM for better convergence. We have added these missing training details in the working copy. Thanks! There are also literature echoing the imperfect LSTM performance in needle searching [1] and decimal addition [2] tasks.
>
> [1] Zhang, Wei, and Bowen Zhou. "Learning to update auto-associative memory in recurrent neural networks for improving sequence memorization." arXiv preprint arXiv:1709.06493 (2017).
>
> [2] Bradbury, James, et al. "Quasi-recurrent neural networks." arXiv preprint arXiv:1611.01576 (2016).

---

> > ### Comment · Reviewer_wwwU · 2024-08-12
> >
> > I thank the authors for addressing my questions. I acknowledge that I have read the reviews posted by other reviewers and the authors' rebuttals. I will raise my score.

---

### Official Review · Reviewer_B9i5 · 2024-07-12

**Soundness:** 3
**Presentation:** 3
**Contribution:** 3
**Rating:** 7
**Confidence:** 4

**Summary:**

**Update after rebuttal:**
My main concerns have been addressed and/or clarified, and some interesting results have been added. I therefore raise my score from 6 to 7, and vote and argue for accepting the paper.

The paper investigates capabilities of randomly initialized, untrained transformers, where only a linear initial and final mapping (embedding and unembedding) are trained. The tasks investigated are: integer addition (with and without modulus), retrieval of a token in the context by specifying the preceding token as a marker, and parenthesis balancing. The paper shows that training only the embedding and unembedding suffices to solve these tasks, implying that untrained transformers have the algorithmic capability to do so. Ablations show that training both, embedding (incl. positional encoding) and unembedding is important to solve all tasks. Further experiments are performed to test capabilities when training only embedding and unembedding: (i) memorization capacity, and (ii) next-token prediction performance trained and evaluated on a dataset of natural language text. In all cases untrained transformers achieve non-trivial performance, but lag behind fully trained models. Finally, the paper tries to identify whether functions implementable by untrained transformers (with a trained linear input- and output-mapping) are limited to low-dimensional linear subspaces or sparse sub-networks - the former with somewhat mixed results but evidence pointing towards an effect, the latter with more consistently negative results (though pruning and knockout-experiments might be needed to be sure). The last experiment of the paper investigates the capability of untrained transformers (only embedding and unembedding are trained) to imitate the input-output mapping of smaller random transformers on random inputs - which works quite well as long as the transformers to imitate are not too large, but overall performance cannot be matched compared to full training of a comparable standard transformer on the task.

**Strengths:**

* Very timely question - while transformers’ inductive biases and implicit simplicity bias have been investigated before, the question of “Which algorithms are there from the beginning, even before training?” is an important piece of the puzzle and it is straightforwardly accessible to experimentation.
* Range of interesting tasks, which can be related to previous results in the literature and span a range of capabilities.
* Important control experiments w.r.t. which parts of the input- and output-mapping need to be trained (answer: neither only the input, nor only the output are sufficient).

**Weaknesses:**

* Perhaps the main weakness is that the paper aims to cover a lot of ground - which means that breadth of experiments is favored over depth. While I appreciate the breadth, and including the attempt to shed some light on the lower-dimensional subspace and sparse subnetwork hypotheses, this does come at the cost of missing some additional experiments and ablations, which in turn means that some of the results and interpretations must be taken with caution or be considered preliminary but not final answers. (see more under ‘Improvements’ below)
* Some of the caveats and alternative explanations that could not be tested in the paper need to be explicitly mentioned (e.g., in the limitations section) and generality of the claims and findings must be explicitly put in relation to these caveats and limitations. I do not mean that the paper intentionally “shoves caveats under the carpet” (far from it), but since these results may receive a lot of attention, laying a solid groundwork for future work that includes pointing out where the foundations need to be strengthened is very valuable.
* (Minor): While I appreciate that the paper is not bloated with overly complex formalism and vacuous math, I think that Section 3 could do with another pass to be slightly more rigorous.
* (Very Minor): there is a fairly large body of literature (including quite a bit of theory) from the 90s and early 2000s on reservoir computing and echo state networks, where the central idea is that a random recurrent network implements virtually any function on the data, and so all that is needed is to train a linear readout while keeping the random recurrent reservoir’s weights frozen (paraphrasing informally, though there are formal function approximation statements for well-defined classes of functions). It would be good to include at least a pointer to this literature (maybe a good survey) in the related work section.

**Verdict:**
The paper addresses a timely and important question with simple, yet insightful and original experiments that, without a doubt, start to fill an important gap in the literature. The paper is generally well-written, though the formalism (Sec. 3) could benefit from slightly more rigor. The main claims in the paper are supported by empirical evidence and the results are interesting and insightful. My main concern with the current manuscript is that it favors breadth of questions over depth. Just the results in Section 4 could easily be expanded into a full paper by adding more ablations and control experiments. Similarly, identifying whether (and how) random transformers implement functions in a lower-dimensional subspace or sparse subnetwork could easily fill a whole publication. Since the paper is pioneering to a large degree, going for breadth to plant several flags is OK, but I think this needs to be supported by a strong discussion of caveats, open questions, and alternative hypotheses that cannot be ruled out yet. I will be concrete in my suggested improvements below. Taking all of this together, I am leaning towards accepting the paper - I do believe it has the potential to become a landmark paper if more work (beyond the rebuttal) was spent, but even in its current state (and with some improvements after rebuttal) its findings will be interesting to a large audience and will spark follow-up work.

**Improvements:**
1. Here is a list of interesting control experiments and ablations that would make the paper stronger. This list is too extensive for the rebuttal phase and I do not expect to see these experiments. But it would be good if the paper could discuss all the unaddressed questions and add them as an explicit caveat to the limitations section w.r.t. the generality of the findings, and when discussing the particular results.
    1. Control experiment: compare against a random LSTM where only the input and output mappings are trained. The paper compares against a fully trained LSTM (which is good), but the ‘Random LSTM’ is missing (this would also be interesting since it relates more directly to the reservoir computing literature). To complicate things, the LSTM may need a wider (or more narrow) hidden state and more layers to solve the task. If done exhaustively, and the current results hold, then the paper could make strong claims regarding the implicit abilities of Random Transformers that LSTMs do not possess, otherwise the observed abilities may apply to neural networks more generally.
    2. Control experiment: replace the transformer with a random MLP - this helps answer how important the attention mechanism is for the algorithmic capabilities tested. The size of the MLP (width and number of layers) may need to differ from the Transformer (maybe having the same overall random parameter count).
    3. Baseline: it would be good to establish a naive baseline difficulty of the tasks in the paper (particularly Sec. 4) and how well they can be solved by having a trainable linear matrix of a certain size. Off the top of my head this could look like replacing the transformer with a single nonlinear layer (with frozen weights) and training the embedding and unembedding - but maybe there is a better baseline for this. If even very simple baselines could solve the tasks by training a linear input and output projection (which I do not expect), then the corresponding tasks would be unsuitable to say much about the algorithmic capabilities of Random Transformers.
2. More caveats:
    1. PCA is linear, which means that it cannot help identify nonlinear lower-dimensional subspaces. There are more sophisticated methods than a PCA, but I think it suffices to just clearly spell out this caveat in Sec. 6 and in the interpretation of the results of that section.
    2. For the neuron basis results in Sec. 6: it may be that one of the main functions of the unembedding is to “select” the outputs of one or more sparse sub-circuits that implement the required functionality. In this case, the activation of other neurons in the transformers is not suppressed and may easily have similar variance. The analysis in Sec. 6, unless I am mistaken, would not be able to separate these highly active but functionally irrelevant circuits from one (or a few) functionally highly relevant sparse subnetwork(s). To do this reliably, pruning experiments may be required. It is fine to not perform these experiments, but I think the results in Sec. 6 do not conclusively rule out with absolute certainty that the algorithms are not implemented via sparse subnetworks, they only show that these subnetworks cannot be identified by checking how much variance the highest-variance neurons explain, which should be added as a limitation.
    3. Fig. 6: though the KL divergence for Random Transformers clearly shows their inferior performance compared to a fully trained full-size Transformer, I am wondering how large that gap is, because it seems that the Random Transformers still achieve non-trivial performance. It would be nice to have a naive baseline in the plot to see how bad the performance of a “bad” model gets (is a KL divergence of 0.8 “quite bad” or actually “still quite good but not optimal”?). A highly related question is whether the KL divergence is dominated by very bad performance on a few datapoints, or just marginally worse across most inputs? The current writing suggests that Random Transformers beyond a target width of 32 fail (more or less) catastrophically.
3. Related work: Questions very related to the paper have spawned a whole research field (reservoir computing or echo state networks) at the end of the last century and I think readers would benefit from a brief pointer to that literature (either a good survey or a small number of important papers in that field). Similarly, certain prompt-tuning techniques can be related to partial training of input embeddings - while a thorough investigation of the connection between prompt tuning and virtually untrained “residue circuitry” in a trained transformer is far beyond the scope of the current paper, I think a sentence or two in the related work or discussion might spark such research.
4. Sec. 3: The mathematical notation is understandable for people familiar with transformers, but could do with another pass to be more rigorous. Particularly important would be to define the dimensionality of the embedding-matrices and unembedding matrices, as this will determine the number of trainable parameters, and how inputs are tokenized.

**Questions:**

**Questions and minor comments:**

1. How are inputs tokenized (L82)? Standard tokenizers are usually optimized to (pre-)compress natural language text, which may skew the results Sec. 5.2, and may make some of the algorithmic problems harder (e.g. by mapping pairs or triplets of integers to tokens which may make the integer addition tasks harder). Ideally there would be an ablation in 5.2 without a tokenizer, but simply discussing this caveat should be fine.
2. 4.1 Tasks: I had the following questions when reading (the answers are in the appendix, but it would be good to have them in the main paper). What is (the range of) $p$ for modular arithmetic tasks? What is (the range of) $k$ for Needle-in-a-Haystack? What is the (range of) length(s) of the decimal numbers? What is the range of input lengths for Parenthesis balancing?
3. How are variable-length inputs dealt with (e.g. in the parenthesis matching task)? Padding with some token and loss masking? Related: what is the context-size of the transformers used?
4. A recent paper investigated transformers’ capabilities across different kinds of algorithmic problems, using the Chomsky hierarchy [Neural Networks and the Chomsky Hierarchy, Deletang et al. 2022]. The main finding was that the algorithmic complexity class was highly predictive of the length-generalization capability of different architectures. It would be nice to state the algorithmic complexity (i.e. where they lie on the Chomsky hierarchy) of the tasks used in the paper.
5. After L98: Unless I missed something this should be minimization of the negative log likelihood. Also $y$ has not been introduced and the notation $y \min x$ may be confusing. Why not use $p(x_{n+1} | x_{1 \ldots n}; E, F, U)$ which was introduced above L89? Maybe just write down the standard (cumulative) log-loss over a sequence of tokens, and then make the difference between full-training and embedding-only training clear by simply stating the arg min over the loss with the respective parameters.
6. L19: missing reference.
7. L128 (nit): While general Dyck recognition requires context-free grammars, parenthesis balancing with a single type of parenthesis should be context-sensitive, right?
8. For Table1 and Table2: state the number of trainable parameters for each setting (either in the table or the appendix).
9. L141-142: “These results thus point toward the role of a transformer-specific inductive bias in the effectiveness of embedding-only training.” to make this statement stronger, random recurrent nets would also need to be investigated, as the inductive biases may be neural-network specific and not just transformer-specific.
10. Reporting only the median in the main paper is good. But it would be nice to show the whole distribution in the appendix (and maybe even a box-plot) to get a clearer picture.
11. 5.1: For the memorization task - please show the learning curves in the appendix. I assume the random transformer and normal transformer use the same training settings: has the random transformer converged at the end of training?
12. Fig. 4: it would be very nice to see another datapoint (ie., a width of 1024) since the cross-entropy loss for the random transformers seems to start to catch up to the normal transformer, which would be an interesting trend.
13. L242: “(but not LSTM sequence models)” - unless I missed something, the paper has not shown results on randomly initialized frozen LSTMs with a trained embedding and unembedding; only fully trained LSTMs.
14. (nit): NeurIPS Checklist, Q4 (Reproducibility): the question specifically asks whether all necessary details for reproducibility are given in the paper regardless of whether code and data are provided. The justification by the authors is: “We will be releasing code and data after some final cleanup.”, which does not address the question.
15. NeurIPS Checklist, Q2 (Limitations): The reference in the justification to the limitations section in the appendix is broken.

**Limitations:**

There is a brief limitations section at the beginning of the appendix. I think it should be expanded by stating limitations regarding the generality of some findings and whether all reasonable alternative explanations can be ruled out given the current experiments and findings. I have listed these under ‘Improvements 1 and 2’.

---

> ### Author Rebuttal · Authors · 2024-08-06
>
> Thanks for the detailed and throughout review! The following are our responses.
>
> > Compare against random LSTM
>
> In this paper, we are trying to narrow our already-pretty-broad discussion to random transformers instead of random neural architecture in general and we have shown that random transformers perform better than fully-trained LSTMs on some tasks, for example. And indeed, from our quick additional experiments it seems random LSTM does need even wider hidden states to solve the algorithmic tasks as 1024-width 2-layer randomly initialized frozen LSTMs (similar setup as transformers: only embedding and unembedding trained) on the needle-in-a-haystack task only reached a (test) accuracy of 16.85%. This is indeed an interesting future direction worth being explored in subsequent works.
>
> > Control experiment; Baseline
>
> While possible, a direct MLP implementation will be unnatural for the variable-lengthed tasks we addressed. As a slight modification, we studied the performance of a modified version of transformer where the attention matrices (specifically the $\text{softmax}(QK^T)$; the V is still used) are placed with lower triangular matrices of 1s. One can also consider this model as a MLP with additional token-mixing prefix sum layers and layer normalizations. We trained such models of 512 width and 2 layers. The result accuracy is given as follows (included in appendix). We can see that such linearized transformers have large performance gaps, confirming the difficulty of our chosen tasks.
>
> - Parenthesis balancing: 97.32%
> - Needle-in-a-haystack: 17.67%
> - Decimal addition: 26.39%
>
> > PCA cannot identify nonlinear lower-dimensional subspaces
>
> Here by subspace we mean linear subspaces and our experiments and analysis are based on that. We have added this as a limitation in the working version.
>
> > Sec. 6 cannot separate active but functionally irrelevant circuits
>
> This is a great point. We have added this as a limitation in the working version.
>
> > Fig. 6: Clarify how large KL divergence gap is
>
> Subjectively a KL of 0.2 is already quite bad. Due to limited space please see the official comment for samples.
>
> > Reservoir computing could be relevant
>
> Thanks for pointing this out! This line of research is indeed very relevant. One tiny but important difference here is that we train both embedding and unembedding in our main settings, which deviates from the purpose of the reservoir computing diagram. We have added a brief introduction to reservoir computing and a few surveys in the related work section.
>
> > Refine mathematical notation in Sec. 3
>
> Thanks for the suggestion! We have refined the writing and added a short paragraph discussing the shape of matrices and parameter counts.
>
> > Clarify tokenization
>
> Tokenization is indeed an important factor in such algorithmic tasks and has been constantly evolving [1]. Due to the scope of the paper, we generally chose the simplest tokenization (e.g. one token per digit in the decimal addition task) in our setups and described the details of our tokenizations in Appendix D. We added the discussion in the limitation section.
>
> [1] Max Buckley. Right to Left (R2L) Integer Tokenization. https://www.beren.io/2024-07-07-Right-to-Left-Integer-Tokenization/
>
> > Specify parameter ranges
>
> Thanks for pointing this out! We added the answers (p=199, at most 30 pairs, 10-digit, at most 60 parentheses) to the main text.
>
> > How are variable-length inputs dealt with? Context size?
>
> Yes, the standard padding and loss masking. The context size generally has the same magnitude as the maximum possible length of the sequence (though usually slightly larger). For example, in the ten-digit decimal addition, we used context size 40. The numbers are also given in the attached code. We believe the result will stay unchanged qualitatively as long as the context lengths are of reasonable magnitudes.
>
> > L98 is off and confusing
>
> Thanks for spotting this out! Following your suggestion we have modified the optimization goal to be $\\arg\\min_{E,U}$ and $\\arg\\min_{E,F,U} \\sum_{x,n\ge 0} -\\log p(x_{n+1} \\mid x_{1\cdots n};E,F,U).$
>
> > L19: missing reference; Broken link in Limitation
>
> Fixed. Thanks!
>
> > Hierarchy of parenthesis balancing
>
> Correct me if I'm wrong, but I think context-free grammar is a special case of context-sensitive grammar. If you mean regular, the language of balanced parentheses is not regular by the pumping lemma.
>
> > Table 1,2: state # of trainable parameters for each setting
>
> Thanks for the suggestion! We have added the parameter counts in Appendix D.2.1. We also attached the table in the rebuttal pdf.
>
> > L141-142: ... a transformer-specific inductive bias in the effectiveness of embedding-only training. Random recurrent nets also need to be investigated.
>
> The “effectiveness” here is transformer-specific as the LSTMs we trained, fully trained or partially frozen, fail to complete the decimal addition task, thus worse performing compared to the random transformer. We do not plan to advocate for other randomly initialized neural architectures including LSTM due to the already pretty broad scope of the paper.
>
> > Show whole distribution
>
> Great suggestion! Changed the plot to a box plot (also attached in the rebuttal pdf).
>
> > Show learning curves for memorization
>
> Yes they converged. Attached the plot to both the appendix and the rebuttal pdf.
>
> > A datapoint of width of 1024 in Fig 4
>
> That is a great suggestion! Unfortunately we are unable to run this experiment due to time and resource constraints. We will include that in the final version if time permits.
>
> > No results on randomly initialized frozen LSTMs with a trained embedding and unembedding; only fully trained
>
> Yes, but here we assumed that randomly initialized LSTMs perform worse than fully trained LSTMs. See the top for additional experiment results.
>
> > Reproducibility
>
> Changed the line to `We tried our best to convey all the experiment details and we will also be releasing the code and data.` Thanks!

---

> > ### Comment · Reviewer_B9i5 · 2024-08-12
> > **Thank you for the detailed clarifications, answers, and additional experiments/results.**
> >
> > The extensive rebuttal has addressed all of my main issues, clarified some of my misunderstandings, and has added significant additional data (that was asked for). I agree with the answers / comments and do not have any further large open issues. I will therefore raise my score to a 7 since I think the paper provides some very interesting insights into capabilities of untrained transformers, and thus indirectly into their inductive biases, which is a very timely and important topic. I would not be surprised if the paper triggers quite a bit of follow-up work.
> >
> > **Minor:**
> > Re "Hierarchy of parenthesis balancing" - ignore my initial comment (not sure what I had in mind when I wrote it). As you correctly say, Dyck languages are context-free (which is a subset of context-sensitive) and parenthesis balancing is non-regular (context-free).

---

> ### Author Response · Authors · 2024-08-06
> **Samples in the circuit imitation task**
>
> > Fig. 6: Clarify how large the KL divergence gap is.
>
> We sampled some inputs (not cherry picked) and the following are the output distributions from the target model and the fully-trained / random transformers. Displayed are the top 5 entries. The KL and TVD at the end of rows are KL divergence and total variational distance (half of L1 distance) from the target distribution to the distribution from the transformers. Subjectively a KL of 0.2 is already quite bad.
>
> ```
> input			 [9, 269, 291, 196, 125, ..., 261, 198, 248, 71, 320]
> 64 width target dist	 261(14.16%) 391(8.30%) 114(5.84%) 140(3.73%) 295(3.23%)
> w128_f0_l1(kl 0.117)	 261(9.97%) 114(5.35%) 391(4.39%) 295(3.79%) 140(3.58%)  		KL=0.07 TVD=0.15
> w512_f0_l3(kl 0.001)	 261(13.78%) 391(8.43%) 114(6.20%) 140(3.57%) 295(3.26%)  		KL=0.00 TVD=0.01
> w512_f1_l1(kl 0.247)	 261(10.77%) 295(5.19%) 27(4.95%) 134(3.60%) 350(2.91%)  		KL=0.17 TVD=0.24
> w512_f1_l3(kl 0.258)	 261(9.16%) 114(5.89%) 295(5.31%) 27(4.65%) 391(3.22%)  		KL=0.14 TVD=0.21
>
> input			 [354, 139, 431, 379, 334, ..., 240, 455, 390, 492, 218]
> 64 width target dist	 393(65.93%) 142(2.90%) 130(1.65%) 379(1.55%) 413(1.29%)
> w128_f0_l1(kl 0.117)	 393(53.46%) 142(3.02%) 413(2.99%) 130(2.28%) 472(2.23%)  		KL=0.08 TVD=0.16
> w512_f0_l3(kl 0.001)	 393(66.30%) 142(2.70%) 379(1.55%) 130(1.54%) 413(1.21%)  		KL=0.00 TVD=0.01
> w512_f1_l1(kl 0.247)	 393(19.74%) 130(5.98%) 413(4.44%) 142(2.10%) 41(2.02%)  		KL=0.65 TVD=0.51
> w512_f1_l3(kl 0.258)	 393(13.00%) 130(7.41%) 413(4.24%) 142(2.43%) 152(2.28%)  		KL=0.90 TVD=0.57
>
> input			 [268, 259, 117, 487, 483, ..., 59, 360, 419, 162, 333]
> 64 width target dist	 500(9.20%) 333(6.67%) 171(4.67%) 462(4.33%) 202(3.98%)
> w128_f0_l1(kl 0.117)	 500(9.89%) 101(5.20%) 39(4.61%) 462(4.23%) 155(3.99%)  		KL=0.17 TVD=0.22
> w512_f0_l3(kl 0.001)	 500(9.89%) 333(7.00%) 171(4.87%) 462(4.20%) 202(4.00%)  		KL=0.00 TVD=0.02
> w512_f1_l1(kl 0.247)	 500(8.47%) 101(4.10%) 462(3.68%) 39(3.21%) 333(3.05%)  		KL=0.14 TVD=0.21
> w512_f1_l3(kl 0.258)	 500(8.00%) 101(5.95%) 39(4.75%) 171(3.70%) 462(3.46%)  		KL=0.19 TVD=0.25
>
> input			 [377, 166, 258, 295, 300, ..., 106, 117, 23, 33, 159]
> 64 width target dist	 401(6.64%) 261(5.30%) 70(4.31%) 219(3.74%) 82(3.48%)
> w128_f0_l1(kl 0.117)	 401(6.73%) 29(5.05%) 261(3.30%) 503(3.04%) 386(2.89%)  		KL=0.15 TVD=0.21
> w512_f0_l3(kl 0.001)	 401(6.84%) 261(5.37%) 70(4.30%) 219(3.73%) 82(3.56%)  			KL=0.00 TVD=0.01
> w512_f1_l1(kl 0.247)	 29(5.72%) 401(5.24%) 261(3.41%) 295(3.16%) 162(2.65%)  		KL=0.20 TVD=0.27
> w512_f1_l3(kl 0.258)	 401(6.22%) 29(5.46%) 295(5.00%) 261(4.02%) 162(2.79%)  		KL=0.25 TVD=0.29
>
> input			 [214, 431, 443, 153, 276, ..., 304, 132, 315, 213, 330]
> 64 width target dist	 393(20.05%) 37(15.42%) 41(10.19%) 428(4.32%) 103(2.44%)
> w128_f0_l1(kl 0.117)	 393(21.93%) 41(12.27%) 37(7.86%) 428(3.89%) 173(2.47%)  		KL=0.09 TVD=0.16
> w512_f0_l3(kl 0.001)	 393(20.26%) 37(15.12%) 41(10.68%) 428(4.15%) 103(2.38%)  		KL=0.00 TVD=0.01
> w512_f1_l1(kl 0.247)	 41(13.53%) 393(11.35%) 37(8.00%) 428(5.61%) 462(2.78%)  		KL=0.19 TVD=0.26
> w512_f1_l3(kl 0.258)	 41(10.51%) 393(10.18%) 37(9.57%) 428(5.00%) 54(3.95%)  		KL=0.21 TVD=0.26
>
> input			 [6, 159, 424, 316, 370, ..., 158, 23, 70, 324, 214]
> 64 width target dist	 70(5.27%) 386(4.85%) 439(4.76%) 29(3.97%) 215(3.96%)
> w128_f0_l1(kl 0.117)	 215(6.72%) 386(6.50%) 70(4.02%) 29(3.61%) 39(2.48%)  			KL=0.11 TVD=0.19
> w512_f0_l3(kl 0.001)	 70(5.25%) 386(4.77%) 439(4.75%) 215(4.03%) 29(3.76%)  			KL=0.00 TVD=0.02
> w512_f1_l1(kl 0.247)	 215(7.09%) 386(7.05%) 357(3.14%) 168(2.50%) 230(2.47%)  		KL=0.25 TVD=0.27
> w512_f1_l3(kl 0.258)	 386(6.14%) 215(4.86%) 29(2.81%) 357(2.61%) 401(2.26%)  		KL=0.25 TVD=0.27
>
> input			 [150, 284, 450, 41, 414, ..., 415, 307, 394, 495, 495]
> 64 width target dist	 130(9.15%) 485(6.04%) 171(4.28%) 261(3.96%) 101(3.64%)
> w128_f0_l1(kl 0.117)	 425(6.22%) 485(5.24%) 130(4.67%) 132(4.13%) 65(3.71%)  		KL=0.11 TVD=0.19
> w512_f0_l3(kl 0.001)	 130(9.75%) 485(5.86%) 261(4.11%) 171(4.07%) 101(3.47%)  		KL=0.00 TVD=0.02
> w512_f1_l1(kl 0.247)	 425(10.05%) 130(6.46%) 132(5.33%) 65(3.84%) 485(2.92%)  		KL=0.23 TVD=0.29
> w512_f1_l3(kl 0.258)	 425(9.15%) 130(7.98%) 132(4.38%) 485(4.15%) 101(2.79%)  		KL=0.16 TVD=0.23
>
> input			 [219, 428, 440, 198, 404, ..., 468, 252, 223, 37, 204]
> 64 width target dist	 41(10.26%) 386(7.45%) 401(6.32%) 496(5.48%) 357(3.59%)
> w128_f0_l1(kl 0.117)	 41(8.47%) 401(7.66%) 386(5.60%) 133(3.48%) 413(2.33%)  		KL=0.10 TVD=0.17
> w512_f0_l3(kl 0.001)	 41(9.87%) 386(7.50%) 401(6.51%) 496(5.64%) 357(3.62%)  		KL=0.00 TVD=0.02
> w512_f1_l1(kl 0.247)	 386(11.59%) 401(7.91%) 41(4.64%) 413(2.83%) 280(2.12%)  		KL=0.34 TVD=0.32
> w512_f1_l3(kl 0.258)	 386(8.80%) 401(6.11%) 41(4.32%) 357(3.04%) 413(2.96%)  		KL=0.34 TVD=0.32
> ```

---

### Author Rebuttal · Authors · 2024-08-06

We would like to thank all the reviewers for their detailed, thorough, insightful, and warm-hearted reviews. Your suggestions and criticisms definitely shaped the paper for better. We are especially pleased to see that most reviewers are generally satisfied with our presentation.

During the rebuttal phase, our efforts can be summarized as follows:

- **Establishing Task Difficulty:** We established the difficulty of the algorithmic tasks both theoretically and empirically. Theoretically, we classified the tasks with the Chomsky Hierarchy. Empirically, we measured performance from a baseline model—a linearized transformer. Please refer to our response to Reviewer B9i5 for more details.
- **Additional Experiment on Random LSTM:** Although our experiments primarily focus on random transformers, we conducted an additional experiment with randomly initialized, frozen LSTMs on the needle-in-a-haystack task. This model, with a 1024-width 2-layer setup, was outperformed by both fully trained LSTMs and random transformers, achieving a test accuracy of only 16.85%.
- **Presentation Refinement:** We improved the clarity and formality of Sections 3 and 6, and added discussions on reservoir computing. In the Appendix, we replaced Fig 7 with a box plot, added parameter counts and hyperparameter details, and included the accuracy curve of training. We also expanded discussions on the limitations of our work.

Please find our detailed responses to individual reviewers below. Once again, thank you for your invaluable feedback!

---

### Decision · Program_Chairs · 2024-09-25

**Decision:**

Accept (poster)

**Comment:**

The paper presents an intriguing exploration of what can be achieved by transformer models that are randomly initialized, with only the input and output embedding layers trained. The authors' finding that such randomly initialized transformers can still perform remarkably well across various tasks is both surprising and thought-provoking. The paper is clearly written, and the results challenge conventional assumptions, contributing valuable insights to the ongoing discourse on the capabilities of transformer models.

However, a critical aspect that the paper currently overlooks is the connection to **neural reprogramming**, which is a significant and relevant area of research. I strongly recommend that the authors thoroughly review the existing literature on neural reprogramming. See [1,2] for instance. Integrating these insights into the discussion would not only strengthen the paper but also ensure that it aligns with current research trends. Without this connection, it is difficult to fully endorse the paper for acceptance. Therefore, I suggest a conditional accept, contingent upon the authors addressing this gap in the final version.

[1] https://arxiv.org/pdf/1806.11146

[2] https://arxiv.org/pdf/2202.10629